# Pyruvate kinase variant of fission yeast tunes carbon metabolism, cell regulation, growth and stress resistance

Stephan Kamrad[1,2,†] , Jan Grossbach[3,†], Maria Rodríguez-López[2], Michael Mülleder[1,4], StJohn Townsend[1,2], Valentina Cappelletti[5], Gorjan Stojanovski[2], Clara Correia-Melo[1], Paola Picotti[5], Andreas Beyer[3,6,*] , Markus Ralser[1,4,**] & Jürg Bähler[2,***]

## Abstract

Cells balance glycolysis with respiration to support their metabolic needs in different environmental or physiological contexts. With abundant glucose, many cells prefer to grow by aerobic glycolysis or fermentation. Using 161 natural isolates of fission yeast, we investigated the genetic basis and phenotypic effects of the fermentation–respiration balance. The laboratory and a few other strains depended more on respiration. This trait was associated with a single nucleotide polymorphism in a conserved region of Pyk1, the sole pyruvate kinase in fission yeast. This variant reduced Pyk1 activity and glycolytic flux. Replacing the "low-activity" *pyk1* allele in the laboratory strain with the "high-activity" allele was sufficient to increase fermentation and decrease respiration. This metabolic rebalancing triggered systems-level adjustments in the transcriptome and proteome and in cellular traits, including increased growth and chronological lifespan but decreased resistance to oxidative stress. Thus, low Pyk1 activity does not lead to a growth advantage but to stress tolerance. The genetic tuning of glycolytic flux may reflect an adaptive trade-off in a species lacking pyruvate kinase isoforms.

**Keywords** cellular ageing; fermentation; glycolysis; oxidative stress; respiration
**Subject Category** Metabolism
**Mol Syst Biol. (2020) 16: e9270**

## Introduction

Inter-linked pathways for carbon metabolism generate both energy in the form of ATP and fulfil key anabolic roles. Organisms tune their carbon metabolism to environmental conditions, including stress or available nutrients, which affects fundamental biological processes such as cell proliferation, stress resistance and ageing (New *et al*, 2014; Valvezan & Manning, 2019). Accordingly, aberrant carbon metabolism is the cause of multiple human diseases (Zanella *et al*, 2005; Wallace & Fan, 2010; Djouadi & Bastin, 2019). Glycolysis converts glucose to pyruvate, which is further processed in alternative pathways; for example, pyruvate can be converted to ethanol (fermentation) or it can be metabolised in mitochondria via the citric acid cycle and oxidative phosphorylation (respiration). Fermentation and respiration are antagonistically regulated in response to glucose or physiological factors (Molenaar *et al*, 2009; Takeda *et al*, 2015). In the presence of glucose, many microbes suppress respiration and grow preferentially by glycolysis, even with oxygen being available. This metabolic state, called aerobic glycolysis (Crabtree, 1929), appears paradoxical, because only full glucose oxidation via the citric acid cycle and respiration will maximise the ATP yield generated per glucose. Aerobic glycolysis, found in Crabtree-positive species, may have been selected because it enables higher rates of ATP production (Pfeiffer & Morley, 2014). Analogously, human cancer cells typically grow by aerobic glycolysis, known as the Warburg effect (Warburg, 1927), thought to increase biosynthetic capacity (Diaz-Ruiz *et al*, 2009; Lunt & Vander Heiden, 2011; Costa & Frezza, 2017). Proposed explanations for how aerobic glycolysis allows faster proliferation involve efficient resource allocation (Basan *et al*, 2015; Mori *et al*, 2019), molecular crowding (Andersen & von Meyenburg, 1980; Zhuang *et al*, 2011; Vazquez & Oltvai, 2016; Szenk *et al*, 2017), an upper limit to the cellular Gibbs energy dissipation rate (Niebel *et al*, 2019), among others (Dai *et al*, 2016; de Alteriis *et al*, 2018; de Groot *et al*, 2019).

1   Molecular Biology of Metabolism Laboratory, The Francis Crick Institute, London, UK
2   Department of Genetics, Evolution & Environment, Institute of Healthy Ageing, University College London, London, UK
3   CECAD, Medical Faculty & Faculty of Mathematics and Natural Sciences, University of Cologne, Cologne, Germany
4   Charité University Medicine, Berlin, Germany
5   Department of Biology, Institute of Molecular Systems Biology, ETH Zurich, Zurich, Switzerland
6   Center for Molecular Medicine Cologne, Cologne, Germany
    *Corresponding author. Tel: +49 22147884429; E-mail: andreas.beyer@uni-koeln.de
    **Corresponding author. Tel: +44 2037962385; E-mail: markus.ralser@crick.ac.uk
    ***Corresponding author. Tel: +44 2031081602; E-mail: j.bahler@ucl.ac.uk
    †These authors contributed equally to this work

Crabtree-positive organisms, including the model yeasts *Saccharomyces cerevisiae* and *Schizosaccharomyces pombe* (Skinner & Lin, 2010), still require some oxygen and basal respiration for optimal cell proliferation (Chan & Roth, 2008). *S. cerevisiae* cells without mitochondrial genome, and thus without respiratory capacity, feature a slow-growth "petite" phenotype (Ephrussi *et al*, 1949). *S. pombe* cannot normally grow without a mitochondrial genome (Haffter & Fox, 1992; Heslot *et al*, 1970; Chiron *et al*, 2007), and blocking oxidative phosphorylation with antimycin A leads to moderate or strong growth inhibition, respectively, in rich or minimal glucose media (Malecki *et al*, 2016). In conditions of low glucose uptake, such as stationary phase, the metabolism of yeast cells is reconfigured towards respiration (DeRisi *et al*, 1997; Zuin *et al*, 2010). Thus, cells tune the balance between respiration and fermentation to meet their metabolic needs (Molenaar *et al*, 2009) in a more nuanced way than captured by qualitative descriptions of aerobic glycolysis.

Given its impact on health and disease, it is important to understand the genetic and regulatory factors that affect cellular carbon metabolism. Here, we investigated the genetic basis and physiological implications for the regulatory balance between fermentation and respiration, using our collection of natural *S. pombe* isolates (Jeffares *et al*, 2015). A few strains featured a higher reliance on respiration during growth on glucose. This trait was associated with a missense variant in pyruvate kinase (PYK). PYK catalyses the final, ATP yielding step of glycolysis, the conversion of phosphoenolpyruvate to pyruvate. PYK can coordinate the activity of central metabolic pathways (Pearce *et al*, 2001; Grüning *et al*, 2011; Yu *et al*, 2018). Most organisms encode several PYK isoforms that are expressed in specific tissues or developmental stages (Allert *et al*, 1991; Muñoz & Ponce, 2003; Bluemlein *et al*, 2011; Israelsen & Vander Heiden, 2015; Bradley *et al*, 2019). Work in budding yeast has implied that the switch from a high- to a low-activity PYK isoform causes increased oxygen uptake, triggering a shift from fermentative to oxidative metabolism (Grüning *et al*, 2011; Yu *et al*, 2018). *S. pombe* possesses only one PYK, Pyk1 (Nairn *et al*, 1995). Exchanging the Pyk1 variant of the standard laboratory strain triggered increased glycolytic flux, which in turn led to substantial adjustments in the metabolome, transcriptome and proteome. These results show that altered PYK activity is self-sufficient to reprogramme metabolism even in the absence of an evolved regulatory signalling system. These findings define a natural metabolic tuning, consisting of a single amino acid change, possibly reflecting an adaptation in a species lacking multiple PYK isoforms. Notably, the standard laboratory strain is among a minority of natural isolates locked in the low-activity state and is thus metabolically and physiologically unusual. These findings highlight the importance of glycolysis in general, and PYK in particular, as a hub in cross-regulating metabolic pathways and coordinating energy metabolism with cell regulation and physiology, including growth and stress resistance.

# Results

## Increased respiration dependence is associated with a missense PYK variant

Treating aerobic glycolysis as a complex, quantitative trait, we assessed the amount of residual respiration on glucose-rich media across a set of genotypically and phenotypically diverse wild *S. pombe* isolates. Resistance to antimycin A, which blocks the respiratory chain by inhibiting ubiquinol-cytochrome c oxidoreductase (Kim *et al*, 1999), was used as a proxy read-out for cellular dependence on oxidative phosphorylation. The standard laboratory strain $972\ h^-$ shows a moderate reduction in maximum growth rate and biomass yield in this condition (Malecki & Bähler, 2016; Malecki *et al*, 2016). We applied a colony-based assay to determine relative fitness of each strain in rich glucose media with and without antimycin A. The resulting resistance scores, i.e. ratios of growth with vs. without antimycin A, showed a large diversity between strains (Fig 1A, Appendix Fig S1, Dataset EV1). Notably, the laboratory strain was among the most sensitive (score = 1, rank = 10 of 154), with the mean score being $1.25 \pm 0.15$ for all strains tested.

The antimycin A resistance trait showed an estimated narrow-sense heritability of 0.54, reflecting the fraction of phenotypic variance explained by additive genetic effects. This heritability was substantially higher than for most of the 223 phenotypes previously reported (Jeffares *et al*, 2015), which indicates a strong genetic basis

**Figure 1. A variant in conserved *pyk1* region is associated with increased sensitivity to antimycin A.**

A Distribution of antimycin A resistance scores for wild isolates, compared to standard laboratory strain (red vertical line). Resistance scores are the ratio of fitness on rich glucose media with vs. without 500 μg/l antimycin A. Fitness was estimated based on colony size on solid media, corrected for spatial and plate effects (Materials and Methods). After quality control, we obtained quantitative fitness scores for 154 strains, with a signal-to-noise ratio of 29.8 and an unexplained variance of 0.12.

B Volcano plot of a genome-wide association using mixed-model linear regression of antimycin A resistance for 118,527 genetic variants. Variants with moderate or high impact are shown in blue. Red dot: the variant at locus I:3,845,516, which causes a T343A change in the Pyk1 amino acid sequence. This variant was among the top scoring (effect size = −0.645, rank 34; *P* = 0.0008, rank 70).

C Boxplot showing antimycin A resistance for 154 strains grouped by the two alleles at the *pyk1* locus. Strains carrying a C at this genomic locus (orange box) generally have higher antimycin resistance than strains carrying the reference allele T (blue box). As is standard, this boxplot and all other boxplots in this manuscript show the median of the data as central line, the quartiles as box and the extend of the rest of the distribution as whiskers. Points which are 1.5 times the inter-quartile range beyond the high and low quartiles are considered outliers and shown individually.

D Sequence alignment of the region of interest in eukaryotic PYK proteins. The threonine residue at the highlighted position 343 is unique for the reference (laboratory) *S. pombe* strains. For this analysis, we collected 25 Pyk1 homologues from a wide variety of eukaryotes, including animals, plants and fungi, using the HomoloGene resource (Accession 37650). We manually expanded the homologous group to include three other species of the *Schizosaccharomyces* genus (Rhind *et al*, 2011). Sequences were aligned with MAFFT (Katoh *et al*, 2017) and visualised with Jalview (Waterhouse *et al*, 2009).

E Phylogenetic tree based on 31 biallelic SNPs in *pyk1* and in 500bp up- and downstream regions. Strains carrying the common A-allele are in orange, while strains carrying the unusual T-allele are in blue (strain names as in ref. Jeffares *et al* (2015)). Strains JB22 and JB50 (underlined and bold) refer to heterothallic and homothallic versions of the laboratory strain, respectively. The pie chart shows the relative allele frequencies at the genomic locus across all wild strains. Genotype calls at the locus were checked manually and were conclusive, i.e. no strain produced reads with both alleles.

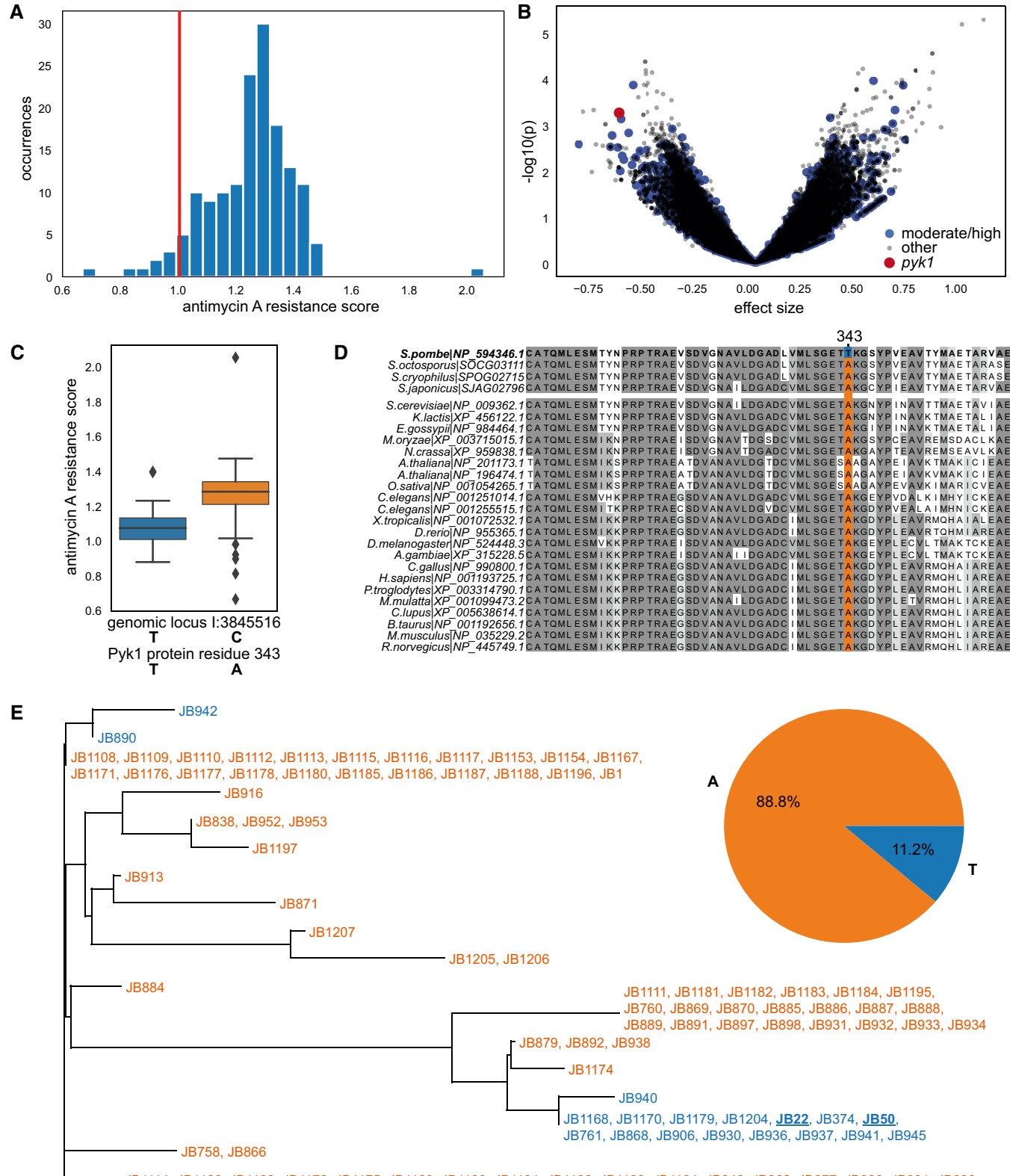

**Figure 1.**

for the dependence on respiration. We performed a genome-wide association study (GWAS) to identify loci linked to resistance among 118,527 small genetic variants, including single nucleotide polymorphisms (SNPs) and small insertions or deletions (Fig 1B). Compared to a recent GWAS in budding yeast (Peter *et al*, 2018), our statistical power was lower, partly due to the low number of

strains and the strong population substructure (Jeffares *et al*, 2015). We therefore manually assessed the associated variants, based on *P*-value, effect size and literature. Among the top-100 associations (by *P*-value, Dataset EV2), eight were predicted to have moderate or high impact as defined by SnpEff (Cingolani *et al*, 2012), and six of those were located in genes with functional annotation in PomBase (Wood *et al*, 2012). Two of these six variants were in *S. pombe* specific genes: *wtf16* and *wtf8* (Hu *et al*, 2017). Other variants were in *ubp9*, encoding a ubiquitin C-terminal hydrolase, in *pfl5*, encoding a cell-surface glycoprotein, and in *jac1*, encoding a mitochondrial 2Fe-2S cluster assembly co-chaperone. One missense variant caught our particular attention: this SNP was among the top scoring (Fig 1B) and leads to a T343A amino acid sequence change in Pyk1, encoding the single PYK in *S. pombe*. Strains with a threonine residue at position 343 ("T-allele") showed a median resistance score of 1.07, while strains with an alanine ("A-allele") showed a ~15% higher median resistance of 1.28 (Fig 1C).

An analysis of 26 PYK protein sequences from diverse eukaryotes revealed strong conservation, showing 45–93% agreement with the consensus sequence called from the alignment (Fig 1D). The T343A mutation was in a region of the protein that is annotated as part of the ADP binding pocket in homologous proteins (Schormann *et al*, 2019). The threonine residue in the reference (laboratory) strain of *S. pombe* was unique in this highly conserved region of all PYK sequences. All other species featured an alanine residue at this position, including three other *Schizosaccharomyces* species (Fig 1D). The reference allele (T-allele) in the laboratory strain was the minor allele in our strain collection, found only in 18 of 161 strains (Fig 1E). The rare T-allele occurred in four unique sequences in the phylogenetic tree, split up over two highly divergent lineages (Fig 1E). This result was confirmed by considering a consensus tree based on the entire genome (Jeffares *et al*, 2015). For 10 of the 18 T-allele strains, the geographical origin is known, with most being isolated in Europe but also one each from Asia and Australia. The substrate is predominantly fermenting grapes (as for most other strains) and one each from lychee and glace syrup, reflecting that most *S. pombe* strains have been isolated from human-created niches. We conclude that the T-allele at position 343 is a rare, naturally occurring allele which appears to have arisen, and been maintained, independently in two distant lineages.

## Replacing *pyk1* allele in laboratory strain leads to higher PYK activity and metabolic adjustment

Many species possess two or more PYK isozymes with different activity and/or expression patterns. In *S. cerevisiae*, the minor isozyme can complement the loss of the major isozyme (Boles *et al*, 1997). The *S. pombe* reference genome (Wood *et al*, 2002), based on the laboratory strain *972*, features only one PYK isoform. To test whether this is also the case for the other strains, we searched the *de novo* assemblies of each wild strain genome (Jeffares *et al*, 2015) for Pyk1 homologues with tblastn (Camacho *et al*, 2009). This search consistently produced a single hit only (Pyk1 itself).

We set out to analyse the effect of the *pyk1* SNP on cellular metabolism. Using seamless CRISPR/Cas9-based gene editing (Rodríguez-López *et al*, 2016), we replaced the *pyk1* T-allele in the laboratory strain for the more common A-allele. We introduced the A-allele in both a heterothallic $h^-$ and a homothallic $h^{90}$ laboratory strain, without any other genetic perturbations. We then selected three independently edited strains for both $h^-$ and $h^{90}$ backgrounds to use as biological replicates throughout this study. Below, we use the abbreviations "T-strain" for the normal laboratory strains and "A-strains" for the edited $pyk1^{T343A}h^-/h^{90}$ strains. These strains allowed us to study the impact of the *pyk1* SNP in a controlled and well-characterised genetic background.

It has been reported that *S. pombe* features lower PYK activity than *S. cerevisiae* (Nairn *et al*, 1995, 1998). However, this conclusion has been derived from the laboratory strain, which contains the unusual *pyk1* allele. To investigate the impact of the *pyk1* SNP on metabolism, we applied a targeted metabolomics workflow based on liquid chromatography–selective reaction monitoring (LC-SRM). We quantified key metabolites potentially affected by PYK activity (Bluemlein *et al*, 2012; Gruning *et al*, 2014), including glycolytic,

---

**Figure 2.   Multi-omic functional investigation of Pyk1 variants.**

A   PCA of metabolite data based on concentrations of 27 central carbon metabolism intermediates. To visualise this high-dimensional data set, we divided the concentration of each metabolite by the median concentration of the T-strain. This normalisation corrects for the large differences in concentrations observed between metabolites but maintains the relative variance within each metabolite. Biological replicates of the T- and A-strains show distinct profiles, largely driven by concentrations of phosphoenolpyruvate, 2-/3-phosphoglyceric acid, NADH and NADPH, as indicated by top loading vectors for each principal component. The biological repeats for the three edited A-strains (circles, squares and triangles) behave similarly, with a variance comparable to that of the biological replicates of the single T-strain.

B   Left boxplot: two glycolytic intermediates directly upstream of PYK were strongly depleted in the A-strain ($n_{T-strain}$ = 9, $n_{A-strain}$ = 8). Right barplot: PYK activity was directly measured using a lactate dehydrogenase-coupled colorimetric enzyme activity assay, showing the mean and standard deviation of substrate conversion rate for three biological replicates, each measured in technical duplicates.

C   Concentrations of fructose-1,6-bisphosphate, which correlate with glycolytic flux, are significantly higher in A-strain (ratio = 1.35, $n_{T-strain}$ = 9, $n_{A-strain}$ = 8).

D   Boxplots for selected metabolomics data indicate differences in energy and redox status between T- and A-strains ($n_{T-strain}$ = 9, $n_{A-strain}$ = 8).

E   Heatmap of the 432 genes that are differentially expressed at the RNA level between the T- and A-strains (FDR < 10%) and are measured in all four conditions (columns). First column: genes ordered by increasing fold-changes for RNAs (computed as $\log_2[T]$-$\log_2[A]$). Second column: fold-changes for proteins (computed as $\log_2[T]$-$\log_2[A]$). Third column: fold-changes for RNAs in cells treated with rapamycin and caffeine (TORC1 inhibition; computed as $\log_2[treatment]$-$\log_2[control]$) (data from ref. Rallis *et al*, 2013). Fourth column: fold-changes for RNAs in cells treated with $H_2O_2$ (oxidative stress; computed as $\log_2[treatment]$-$\log_2[control]$) (data from Chen *et al*, 2003). $\log_2$ fold-changes are capped at absolute values of 1 for all columns.

F   Gene Ontology (GO) enrichment analysis of differentially expressed transcripts (using all measured transcripts as background). Shown are Biological Process terms only, with additional plots for Molecular Function and Cellular Component terms in Appendix Fig S5. The length of the bar represents the significance of the enrichment, while its colour reflects the direction and magnitude of differential expression (A- vs. T-strain) of individual genes annotated to this term.

Data information: Significance keys: *$P$ < 0.05, **$P$ < 0.005, ***$P$ < 0.0005 (Welch's *t*-test).

---

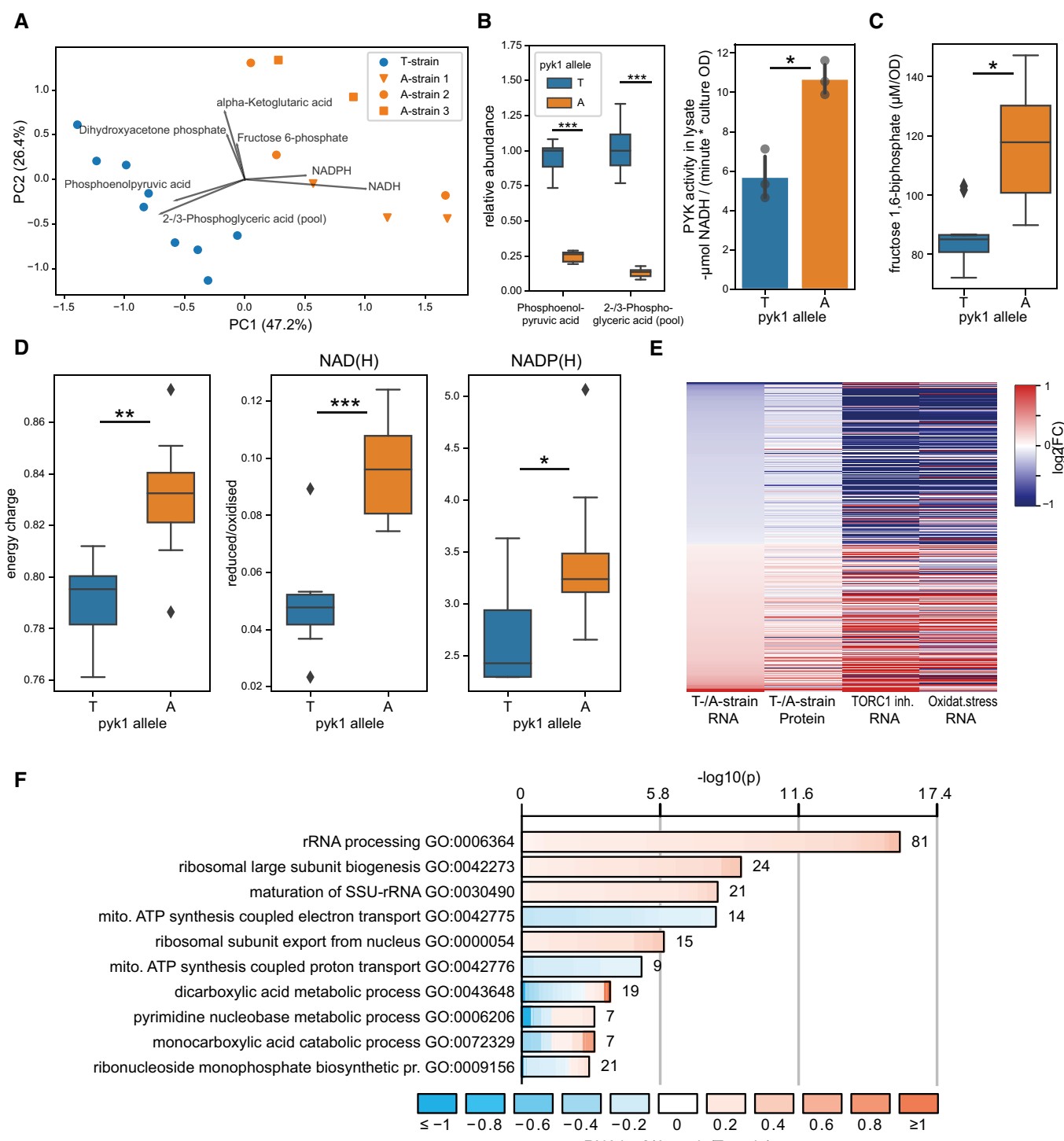

Figure 2.

pentose-phosphate pathway and citric acid cycle intermediates as well as redox cofactors and adenine nucleotides. Data were obtained for nine replicates of the T-strain and eight replicates of the A-strain (Figs 2A–D and 3, Appendix Figs S2 and S3, Dataset EV3). Technical replicates sampled from the same culture, but prepared and measured separately, were highly correlated ($r_{Pearson}$ = 0.93), indicating that our workflow was robust and that most of the observed

variation was biological. Over the entire data set, the median coefficient of variation was 17.2%. A principal component analysis (PCA) of metabolite data distinguished all strains based on the SNP in *pyk1* (Fig 2A).

We observed a strong depletion of glycolytic intermediates upstream of PYK, with mean levels of phosphoenolpyruvate and 2-/3-phosphoglyceric acid in the A-strains only 25.9% and 12.7% of

those in the T-strain (Fig 2B, $P_{adj} = 8.7 \times 10^{-8}$ and $3.7 \times 10^{-6}$, Welch's *t*-test, Benjamini–Hochberg-corrected, *t*-tests are unpaired and two-sided throughout). This result suggests that the A343T mutation reduces the activity of the Pyk1 enzyme. We directly tested this hypothesis by determining PYK activity in lysates of 3 biological replicates each of the A- and T-strain grown in rich glucose media, using a lactate dehydrogenase-coupled photometric assay, in technical duplicates. With a buffer composition similar to those previously used (Gehrig *et al*, 2017), the A-strains showed a 87.3% higher activity compared to the T-strain ($P = 0.0072$, Welch's *t*-test) (Fig 2B, Appendix Fig S4).

The metabolomics data allowed us to also estimate several physiological parameters. The A-strains exhibited significantly higher levels of the flux-signalling metabolite fructose 1,6-bisphosphate (ratio 1.35, $P_{adj} = 0.015$; (Litsios *et al*, 2018; Fig 2C). The levels of this metabolite strongly correlate with glycolytic flux in several yeast species (Christen & Sauer, 2011; Huberts *et al*, 2012), consistent with a higher flux in the A-strain. Furthermore, cellular energy charge was 4.7% higher in the A-strains (Fig 2D; $P = 0.002$, Welch's *t*-test; Atkinson & Walton, 1967). These values were within the range reported for other organisms (De la Fuente *et al*, 2014). We used the ratio of the reduced to oxidised forms of NAD(H), NADP(H) and L-glutathione as read-outs for cellular redox status (Fig 2D). For NAD(H), the A-strains showed an increase from 0.048 to 0.096 ($P = 0.0001$, Welch's *t*-test). The same pattern was evident for NADP(H), where the median ratio was 3.24 in the A-strains, but only 2.43 in the T-strain ($P = 0.024$). Data regarding the oxidation state of glutathione were not entirely conclusive: the A-strain showed a significantly higher concentration of the reduced isoform of L-glutathione (ratio 1.30, $P_{adj} = 0.015$, Welch's *t*-test; Appendix Fig S3, Dataset EV3), but no significant difference was apparent in the ratio of the reduced to oxidised isoforms, where the median for the A-strains was 2.01 vs. 1.89 for the T-strain ($P = 0.289$, Welch's *t*-test). Overall, these results are in line with the paradigm that NADH/NAD$^+$ ratios are maintained at low levels to maximise availability of electron acceptors for catabolic processes, while NADPH/NADP$^+$ ratios are maintained at high levels to provide electrons for anabolic processes and the antioxidant response (Blacker & Duchen, 2016). As part of the antioxidant defence, which includes glutathione-, peroxiredoxin- and thioredoxin-dependent reduction systems, NADPH is limiting when cells are challenged with oxidative stress (Carmel-Harel & Storz, 2000; Drakulic *et al*, 2005; Vivancos *et al*, 2006; Veal *et al*, 2014). While analytical methods cannot distinguish between different compartments or sub-populations of these cofactors (Sun *et al*, 2012), and our sample extraction method may allow some interconversion between reduced and oxidised isoforms (Lu *et al*, 2018), these findings are consistent with the hypothesis that the A-strain respires less and thus has a lower oxidative burden.

### Increased PYK activity leads to transcriptome and proteome changes reflecting increased fermentation and decreased respiration

To further analyse the effects of the *pyk1* SNP, we characterised the transcriptomes and proteomes of the T- and A-strains using RNAseq and mass spectrometry. We could quantify 7,750 transcripts (including non-coding RNAs) and 3,234 proteins in both strains (Dataset EV4), allowing for a broad analysis of genome regulation. The expression of *pyk1* itself was similar in the T- and A-strains at both the transcript ($\log_2[$fold change$] = 0.016$, $P_{adj} = 0.91$) and protein level ($\log_2[$fold change$] = -0.001$, $P_{adj} = 0.94$), indicating that the differences between the two strains were not caused by changes in *pyk1* expression. Notably, the *pyk1* allele replacement led to substantial changes in both the transcriptome and proteome. Overall, 960 transcripts and 434 proteins were differentially expressed between the T- and A-strains, at a false discovery rate (FDR) of $\leq 10\%$. While changes at the transcriptome and proteome levels generally correlated well ($r = 0.65$ for all genes with differentially expressed transcripts and/or proteins), we also found a large number of genes to be regulated exclusively at the protein level (Appendix Fig S7). These proteins were enriched in functions related to cytoplasmic translation and depleted in functions related to ribosome biogenesis (Appendix Tables S1 and S2). This result raises the possibility that post-transcriptional gene regulation plays an important role in controlling translation in this case.

The differentially expressed genes were enriched in functions related to respiration and energy-demanding processes, like translation and ribosome biogenesis. These enrichments were evident at both the level of the transcriptome (Fig 2F, Appendix Fig S5) and proteome (Appendix Fig S6). Transcripts encoding respiratory chain and oxidative phosphorylation proteins were more highly expressed in the T-strain (Fig 3), while those related to ribosome biogenesis and rRNA processing were more highly expressed in the A-strain. Some functional terms (e.g. NAD-binding) contained genes that were strongly regulated in either direction. Several of the most differentially expressed transcripts and proteins were directly involved in pyruvate metabolism (Fig 3). The *mae2* gene was most strongly regulated at both transcript and protein levels, being more highly expressed in the T-strain. Mae2 is an enzyme that catalyses the reaction from malate and oxaloacetate to pyruvate (Viljoen *et al*, 1994). Thus, the T-strain may up-regulate Mae2 to replenish pyruvate using an alternate way that is largely independent of glycolytic flux; alternatively, Mae2 could function as anaplerotic enzyme. The *pdc101* and *atd1* genes, on the other hand, were expressed more highly in the A-strain. These genes encode pyruvate decarboxylase and aldehyde dehydrogenase, respectively, and their induction is consistent with higher glycolytic flux and increased fermentation (Malecki *et al*, 2016). These expression changes are consistent with the central role of PYK in glycolysis and the observed metabolic effects mediated by the Pyk1 variants.

The target of rapamycin complex 1 (TORC1) signalling pathway controls carbon metabolism and promotes aerobic glycolysis in response to cellular nutrients (Valvezan & Manning, 2019). Enrichment analysis using AnGeLi (Bitton *et al*, 2015) revealed substantial overlaps between the differential expression signature of the T- and A-strains and the signature of TORC1 inhibition (Fig 2E; Rallis *et al*, 2013): 58 transcripts and 33 proteins induced by TORC1 inhibition were more highly expressed in the T-strain ($P = 6.1 \times 10^{-12}$ and $4.1 \times 10^{-7}$, respectively), while 118 transcripts and 72 proteins repressed by TORC1 inhibition were more lowly expressed in the T-strain ($P = 1.8 \times 10^{-59}$ and $2.1 \times 10^{-30}$, respectively). Thus, the expression signature of the T-strain resembles the signature caused by TORC1 inhibition, which leads to reduced glycolysis. Moreover, genes induced in response to oxidative stress were also differentially expressed; examples include *gpx1*, encoding glutathione peroxidase,

and *grx1*, encoding glutaredoxin. Accordingly, the differential expression signature between the T- and A-strains also showed substantial overlaps with the core environmental stress response triggered by oxidants and other stresses (Fig 2E; Chen *et al*, 2003): 63 transcripts induced by oxidative stress were higher expressed in the T-strain ($P = 7.0 \times 10^{-4}$), while 149 transcripts and 94 proteins

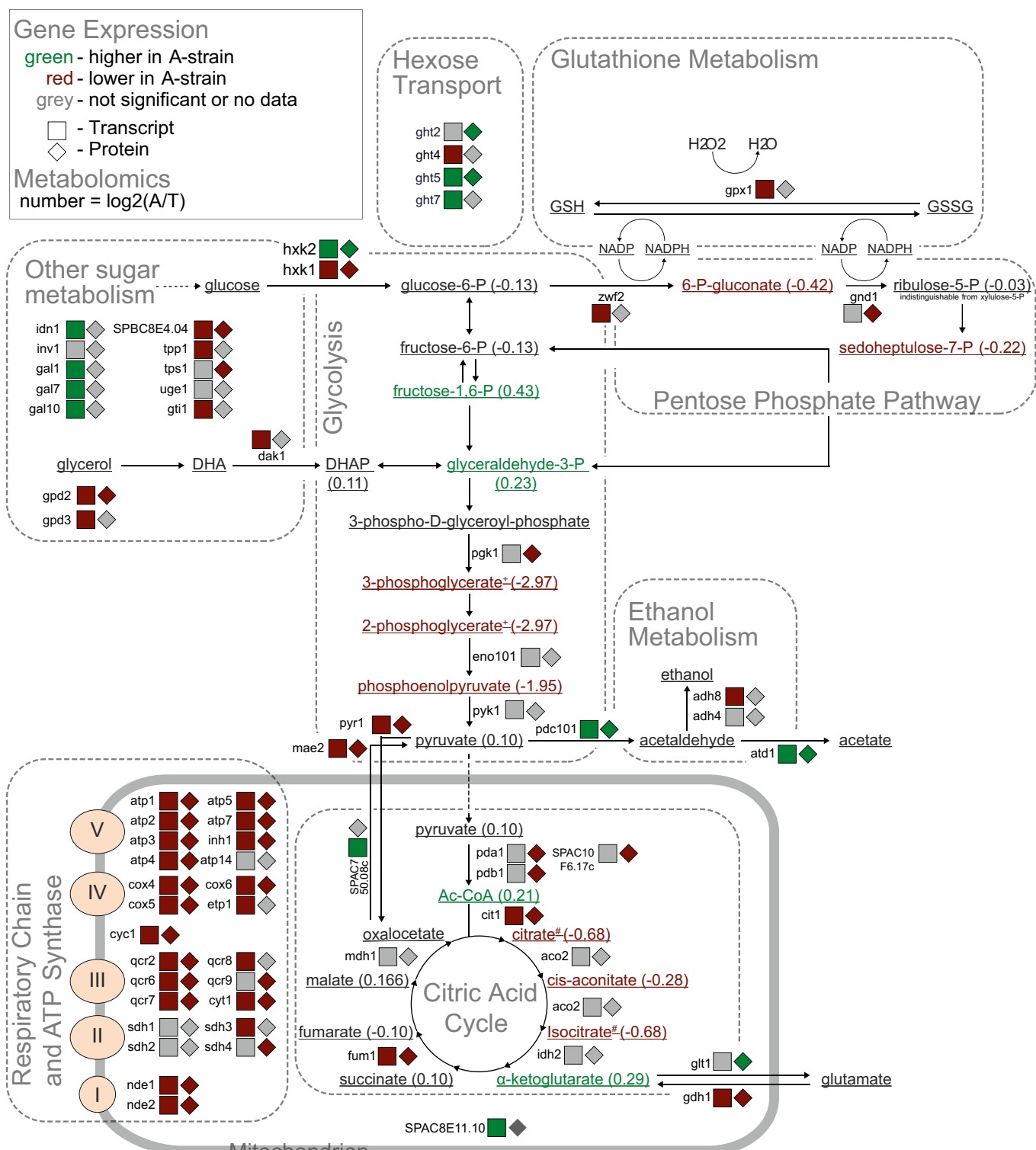

**Figure 3. Metabolic network showing metabolome, transcriptome and proteome changes triggered by *pyk1* allele replacement.**

Metabolites, transcripts and proteins are coloured by their abundance ratios in A-strain relative to T-strain (see legend top left for details). For clarity, enzymes without protein or transcript changes are not shown (e.g. most upper glycolysis). Two metabolite pairs are indistinguishable by our LC-MS method (marked as +" and #").

repressed by stress were lower expressed in the T-strain ($P = 2.8 \times 10^{-68}$ and $1.4 \times 10^{-37}$), respectively. Thus, the expression signature of the T-strain also resembles the general signature of cells exposed to different types of stress, likely reflecting the higher load of reactive oxygen species from respiration. Together, the observed gene-expression reprogramming suggests that the T-strain features higher respiration and pentose-phosphate pathway activity (Fig 3). Similarly, budding yeast strains genetically engineered to alter PYK activity reconfigure their metabolism, with a reduced activity leading to higher respiration (to meet energy demands) and pentose-phosphate metabolism (to increase reducing agents required to detoxify reactive oxygen species produced by respiration; Grüning *et al*, 2011).

## Increased glycolytic flux increases cellular growth and glucose uptake but decreases oxygen consumption and biomass yield

Given the metabolic and gene-expression changes associated with the *pyk1* allele replacement, we expected that the A-strain will feature phenotypic changes at the cellular level, in particular increased growth. We grew the A- and T-strains in biological triplicates in rich media and measured growth by optical density at 600 nm for 6 h of exponential growth. As expected, the A-strain grew more rapidly than the T-strain (Fig 4A). Growth rates were calculated by fitting a line to $\log_2$-transformed data (Fig 4B). The doubling times were 1.94 and 1.85 h for the T- and A-strains, respectively, a 4.7% decrease. Our metabolomics and gene-expression data suggested that this faster growth rate is due to a shift in the fermentation/respiration balance towards fermentation. Accordingly, we detected less residual glucose in A-strain cultures after 8 h of exponential growth, normalised to the final OD (Fig 4C, $P = 0.04$, Welch's *t*-test). Moreover, the A-strain consumed oxygen at a 35% lower rate than the T-strain (Fig 4D). Since energy production from respiration is more efficient than from fermentation, we expected a lower final biomass in the A-strain. Indeed, the final biomass, reported as the ratio of dry biomass to glucose in the fresh media, was $10.38 \pm 0.14\%$ for the A-strain compared with $10.88 \pm 0.19\%$ for the T-strain ($P = 0.02$, Welch's *t*-test; Fig 4E). The final biomass, however, could potentially be confounded by growth phases using other carbon sources, previously produced by the cells or found in the media. Overall, we conclude that the SNP in *pyk1* has a pronounced inhibitory effect on cell proliferation in the standard laboratory strain under a standard growth condition.

## Increased glycolytic flux modulates stress resistance and chronological lifespan

We next investigated which other cellular phenotypes are affected by this change in glycolytic flux. We first confirmed that the T-strain was more sensitive to inhibition of respiration by antimycin A than the A-strain (Fig 5A), which supports the prediction from the GWAS. We hypothesised that a natural SNP in a key metabolic enzyme could differentially affect fitness on different carbon sources. While the A-strain grew more rapidly on glucose, the T-strain might have fitness advantages in other conditions. To test for such a trade-off, we examined 12 common carbon sources in four different base media (Fig 5B, Dataset EV5). Both strains showed rapid cell growth on glucose, fructose and sucrose, intermediate

growth on raffinose, mannose and maltose, and slow growth on the other carbon sources. Consistent with the result in Fig 4, the A-strain grew faster than the T-strain on the fermentable carbon sources glucose, fructose and sucrose. In the other carbon sources, the two strains showed similar growth. Thus, the T-strain did not show increased fitness in any of our conditions. We also tested for differential growth of the A- and T-strains on different nitrogen sources. Both strains showed substantial growth on 54 of the 95 nitrogen sources (Fig 5C, Dataset EV6). The A-strain grew about twofold better than the T-strain on L-phenylalanine but worse on L-cysteine. Validation by spot assays on solid media, however, could only confirm the difference for phenylalanine (Fig 5E). In conclusion, these broad phenotypic assays did not support the idea that the T-allele might represent an adaptation to specific carbon or nitrogen sources.

Only a fraction of natural environments might enable rapid proliferation as in the laboratory. Thus, resistance to stress could be a more important selection factor in determining fitness. Trade-offs are a key concept in evolutionary adaptation (Ferenci, 2016), and microbes show an anti-correlation between growth rate and stress resistance (López-Maury *et al*, 2008; Zakrzewska *et al*, 2011). In budding yeast, artificially reduced glycolytic flux leads to increased resistance to oxidative stress (Grüning *et al*, 2011), and mammalian cells show a similar feature (Anastasiou *et al*, 2011). We therefore assessed the ability of the A- and T- strains to endure oxidative stress triggered by hydrogen peroxide ($H_2O_2$) or diamide. Indeed, the T-strain was substantially more resistant to both oxidants than the A-strain (Fig 5A). Oxidative stress is a by-product of cellular respiration, and the T-strain may feature a higher basal protection from oxidative stress due to higher respiratory activity. This protection is consistent with our observation that core environmental stress response genes were more highly expressed in the T-strain. The environment can also be a source of oxidants, e.g. in microbial communities with $H_2O_2$-producing lactic acid bacteria (Ito *et al*, 2003; Ponomarova *et al*, 2017). Thus, a natural SNP promoting oxidative stress resistance may be beneficial. Our observation that lower glycolytic flux increases oxidative stress resistance, via higher pentose-phosphate pathway flux, is in line with previous studies in *S. cerevisiae*, where glycolytic flux has been reduced by mutations in triosephosphate isomerase (Ralser *et al*, 2007; Gruning *et al*, 2014). We conclude that the slower growth of the T-strain, compared to the A-strain, is offset by an increased resistance to oxidative stress. Both traits may be systemic properties emerging from the up-regulation of respiration at the cost of fermentation, triggered by the T-allele.

We wondered whether the T-strain might feature fitness advantages in stress conditions other than oxidative stress. To this end, we screened for differential growth of the A- and T-strains on 72 different drugs and toxins. The strains appeared to be differentially sensitive to nine compounds (Fig 5D). The A-strain was more resistant to barium chloride, D-L-alanine hydroxamate, caffeine (pleiotropic effects, including TOR inhibition, Rallis *et al*, 2013), chlorpromazine (causes membrane stress, De Filippi *et al*, 2007), capreomycin (binds to ribosomes, Lin *et al*, 2014) and phenylarsine oxide (inhibitor of tyrosine phosphatases, Oustrin *et al*, 1995). Notably, the A-strain was also more resistant to thallium (I) acetate, which is highly toxic due to its similarity to potassium ions and binds to mammalian PYK with a stronger affinity, but weaker

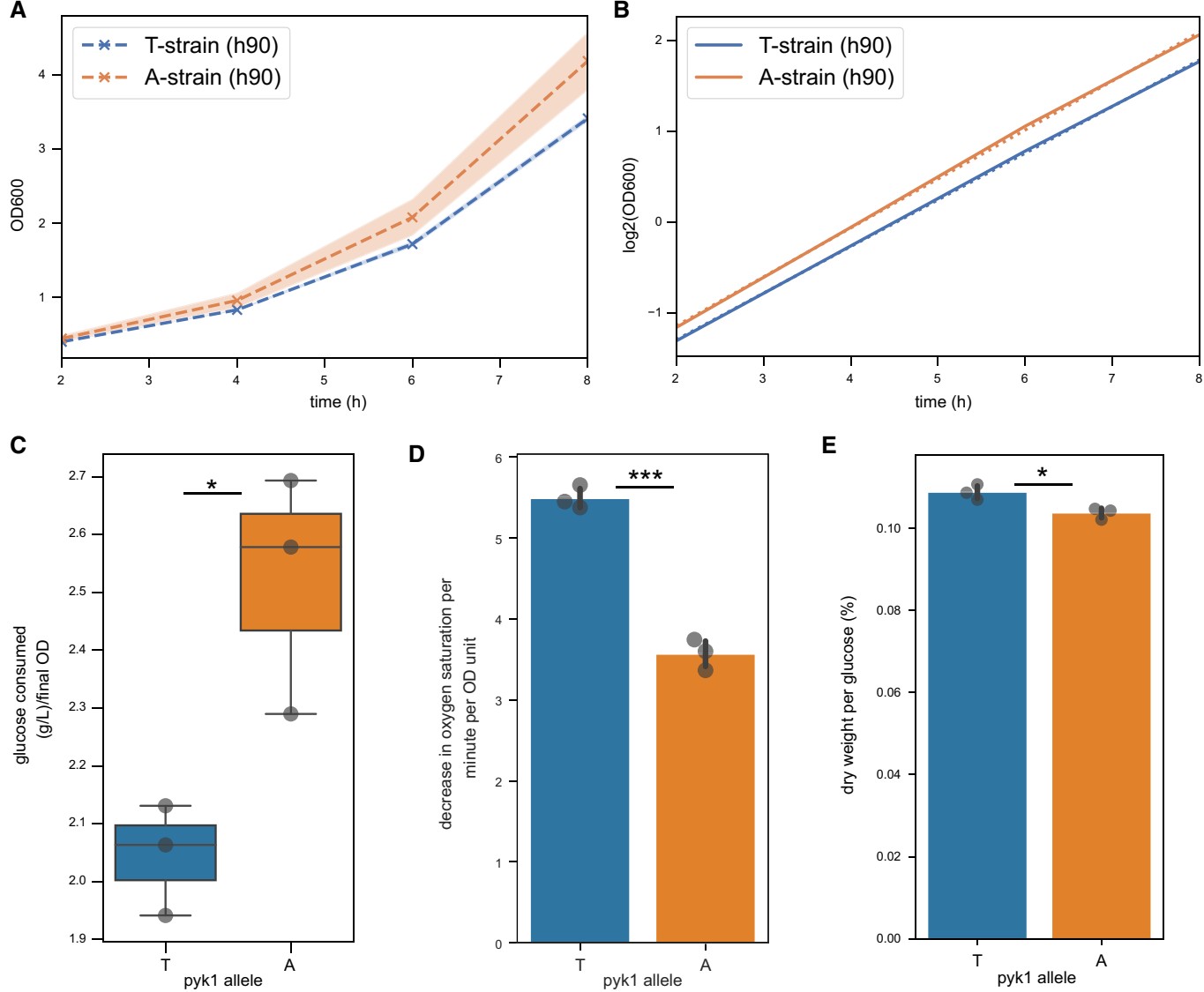

**Figure 4. Physiological characterisation of pyk1 variant.**

A  Growth measured by optical density (OD) at 600 nm for 3 biological replicates each of the T- and A-strains over 6 hrs of exponential growth in YES media. The x-axis refers to the time since inoculation of the culture.

B  The same data as in (A) plotted on a $\log_2$ scale (solid lines) with lines fitted (dotted lines). The doubling time is the inverse of the slope.

C  Media samples were taken from the same cultures (n = 3 for both strains) at the 8 hr timepoint and the remaining glucose was quantified. The consumed glucose was calculated based on the amount of glucose measured in the same, fresh media and normalised to the OD of each culture at the time of sampling.

D  Oxygen consumption rates in A- and T-strains (ratio of means = 0.65, P = 0.0002, Welch's t-test, three biological replicates, each measured in technical duplicates).

E  Culture dry weight after 24 h was measured reported as a fraction of the weight of glucose put into the media for three biological replicates per strain.

Data information: Vertical bars show the mean of the data. Error bars and shaded areas in all cases denote standard deviation. Significance keys: *$P$ < 0.05, ***$P$ < 0.0005 (Welch's t-test).

activating effect, than potassium (Kayne, 1971; Reuben & Kayne, 1971). The T-strain, on the other hand, was more resistant to acriflavine, an antiseptic, and to EGTA, a chelator of bivalent cations (including $Mg^{2+}$ which activates *S. cerevisiae* PYK, Rhodes *et al*, 1986). We manually inspected dose–response curves (Appendix Fig S8) to select compounds for validation based on overall difference in maximum growth rate and consistency across concentrations. Using spot assays, we could validate the differential sensitivity to

four selected compounds: caffeine, phenylarsine oxide, EGTA and chlorpromazine (Fig 5E). It is striking that the *pyk1* SNP differentially affected the resistance to this broad range of stresses, suggesting a general role of glycolysis in stress resistance beyond the known role in oxidative stress.

The existence of a fundamental link between metabolism and lifespan is well known, e.g. through the observation that dietary restriction extends lifespan from yeast to humans (Al-Regaiey,

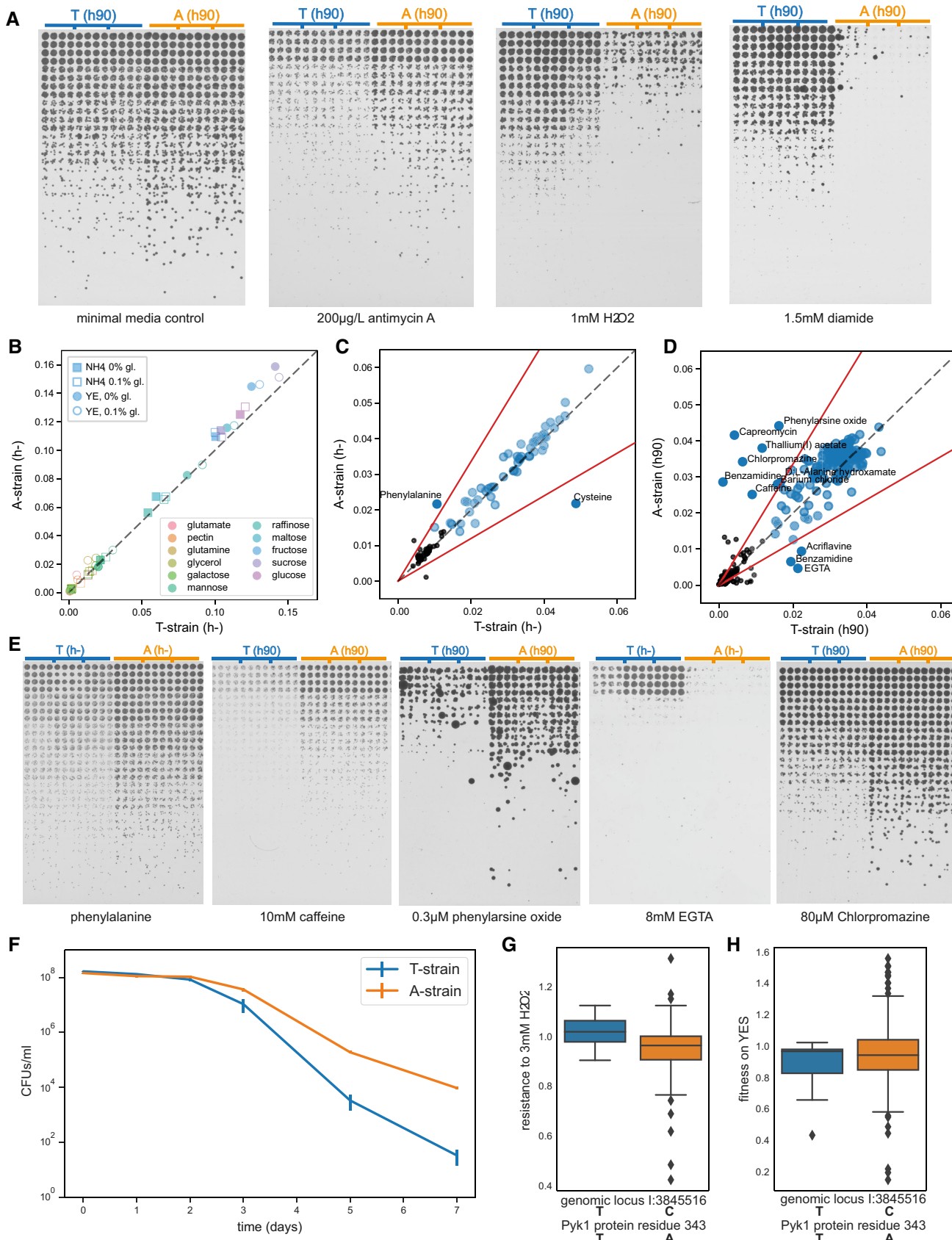

**Figure 5.**

**Figure 5. Cellular phenotypes mediated by *pyk1* allele replacement.**

A  Spot assays on solid media from a threefold dilution series of exponential cultures at the same cell density in 96-well plates (3 biological replicates of each strain) and spotted in 16 technical replicates (each dilution in 4 × 4 square). The A-strain is more resistant to antimycin A but less resistant to oxidative stress triggered by $H_2O_2$ or diamide. A control without toxin (left) was included in each batch of spot assays performed and a representative image is shown here.

B  Fitness (approximated by maximum slope of smoothed growth curves) of A- and T-strains on 12 carbon sources, with either yeast extract (YE) or ammonium ($NH_4$), with or without 0.1% priming glucose to support initial growth. For all 48 conditions, two biological replicates of A- and T-strains were grown in technical quadruplicates each. Dotted lines in panels B-D mark a fitness ratio of 1 (i.e. same fitness).

C  Fitness of A- and T-strains on 95 nitrogen sources on Biolog Phenotype MicroArrays. Conditions with no substantial growth were excluded (black circles, maximum slope < 0.015). Red lines show arbitrary significance cut-off, put at $|\log_2(\text{A-strain/T-strain})| > 0.75$.

D  Fitness of A- and T-strains on 72 different drugs and toxins, at 4 concentrations each, on Biolog Phenotype MicroArrays. Graph details and cut-off as in (C). (Results for benzamidine were inconclusive, with both strains comparatively resistant in one concentration each.)

E  Spot assays as in (A) to validate selected results from (C) and (D). Several assays were performed with both h- and h90 strains, and no mating-type-specific differences were evident between the two sets of allele swap strains.

F  Chronological lifespan of A- and T-strains, i.e. the proportion of non-dividing cells in stationary phase that maintain proliferative potential after refeeding. The data show colony forming units (CFUs) per ml of culture over 7 days of stationary phase in glucose-depleted rich media. Three biological repeats were carried out for both strains, with each repeat measured as technical triplicates. Error bars represent standard error of the biological replicates.

G  Boxplot showing resistance to 3 mM $H_2O_2$ grouped by *pyk1* allele for 156 strains from our collection. The T-strains had a higher mean fitness in $H_2O_2$ than the A-strains ($1.01 \pm 0.06$ vs. $0.95 \pm 0.11$; $P = 0.0021$, Welch's *t*-test). The resistance score was obtained as for antimycin A.

H  Boxplot showing growth fitness on rich media, grouped by *pyk1* allele for 158 *S. pombe* strains ($0.88 \pm 0.16$ vs. $0.94 \pm 0.24$; $P = 0.18$, Welch's *t*-test).

2016). In various model systems, increased respiration (Bonawitz *et al*, 2007; Roux *et al*, 2009; Zuin *et al*, 2010; Pan *et al*, 2011) and slower growth (Yang *et al*, 2011; Rallis *et al*, 2014; Janssens & Veenhoff, 2016; Smith *et al*, 2018) correlate with increased lifespan. We therefore expected the T-strain to be longer-lived than the A-strain and measured the chronological lifespan of both strains. Surprisingly, the A-strain was longer-lived, with a mean viability of 25.3% after 3 days, compared to 6.5% for the T-strain (Fig 5F). Reduced glycolytic flux has been reported to shorten replicative lifespan in budding yeast (Ralser *et al*, 2007). We therefore speculate that unrestricted glycolytic flux generally promotes longevity, but mechanistic processes involved will require further investigation.

Our findings show that the *pyk1* SNP has substantial effects on growth rate and oxidative stress resistance in the genetic background of the standard laboratory strain. Are such effects generally evident in other strain backgrounds? To address this question, we measured oxidative stress resistance and growth rates for all wild strains. Indeed, the T-allele was significantly associated with higher resistance to $H_2O_2$ (Fig 5G). With respect to growth rate, on the other hand, no significant difference was evident between strains containing the T- or A-alleles (Fig 5H). These results suggest that the *pyk1* SNP can play a substantial role in oxidative stress resistance, while growth rate may be a more complex trait, which is controlled by many other loci or buffered by counteracting mutations.

## Discussion

Many eukaryotes, from budding yeast to humans, possess low- and high-activity PYK isozymes. PYK activity has been implicated in coordinating fermentation with respiration in synthetic *S. cerevisiae* models (Pearce *et al*, 2001; Grüning *et al*, 2011). The low-activity isoform of *S. cerevisiae* is expressed under respiratory growth conditions (Boles *et al*, 1997). Using an unbiased, genome-wide approach, we identified a naturally occurring SNP in the sole *S. pombe* PYK gene. Our findings show that this SNP affects PYK activity, possibly by impairing substrate binding, and glycolytic flux which is sufficient to cause a shift in the respiration–

fermentation balance. When we replaced the T-allele of the laboratory strain with the common A-allele, which is broadly conserved in most *S. pombe* strains and in all other eukaryotes examined, glycolytic flux increased and oxygen consumption decreased. This metabolic adjustment led to changes in gene expression at the transcriptome and proteome levels, resembling the signatures of rapidly proliferating cells with high TORC1 activity and no stress exposure. At the cellular level, the allele replacement led to increased growth and chronological lifespan but decreased resistance to oxidative stress. Cellular growth and stress resistance are linked with gene regulation, although cause–effect relationships are poorly understood (López-Maury *et al*, 2008; Morano *et al*, 2012; Pir *et al*, 2012; Slavov & Botstein, 2013; Tamari *et al*, 2014; Hesketh *et al*, 2019).

The extraordinary plasticity in response to altered glycolytic flux, triggered by a single nucleotide change, highlights the fundamental impact of glycolysis on cellular control, physiology and adaptation. Possessing only a single PYK isoform, *S. pombe* is unlikely to have a pre-existing genetic or signalling programme for the regulation of high- and low-activity PYK states. Yet, a mutation in Pyk1 that changes its activity is sufficient to induce coherent metabolic, regulatory and cellular responses. A PYK-induced change in glycolytic flux is hence the cause, not a consequence, of major changes in cellular metabolism, regulation and physiology. The finding that a new metabolic programme can be triggered by an intracellular cue is consistent with a report showing that overexpression of one transcription factor in *Komagataella phaffii* is sufficient to turn this Crabtree-negative yeast into a Crabtree-positive one (Ata *et al*, 2018). These findings support the idea that a flux-sensing mechanism could regulate the balance between respiration and fermentation (Huberts *et al*, 2012).

What might be the evolutionary and ecological role of the *pyk1* SNP? We propose that the mutation in the laboratory strain is beneficial given its maintenance at a strongly conserved position, its occurrence in two independent *S. pombe* lineages, its associated phenotypes and the use of low-activity isoforms in other organisms. The literature suggests that low PYK activity could help cells to retain more carbon intermediates for biosynthesis (Christofk *et al*, 2008; Lunt *et al*, 2015; Allen & Locasale, 2018). Our results, and

other recent research (Morita *et al*, 2018), do not support this hypothesis as cells with low Pyk1 activity grow slower and form less biomass. Our phenotypic assays also do not support the possibility that the *pyk1* SNP is adaptive on specific carbon or nitrogen sources. However, we have identified several stress conditions where the laboratory strain exhibits higher fitness than the allele replacement strain, most notably oxidative stress. It is plausible that stress resistance has provided the selection factor for the low-activity Pyk1 allele. Accordingly, we propose that altered stress tolerance provides a biological rationale for the evolution of systems that allow conditional switching between high- and low-activity PYK isozymes.

# Materials and Methods

### Wild strain phenotyping and GWAS

We constructed two arrays of 384 strains, each containing a reference grid of 96 JB22 colonies (standard laboratory strain 972), around which 159 wild isolates from our collection (Jeffares *et al*, 2015) were randomly arranged in triplicates, with additional internal, interspersed JB22 controls. A RoToR HDA pinning robot (Singer Instruments) was used to copy the arrayed strains onto various growth media. Plates were grown for 2 days at 32°C, and images were acquired by transmission scanning (Epson V800 Photo). Colony sizes were determined with *gitter* (Wagih & Parts, 2014) and corrected for spatial biases using reference-grid normalisation (Zackrisson *et al*, 2016), as implemented in our freely available pipeline (preprint: Kamrad *et al*, 2020). Further, strains which did not grow at all (colony size < 10 pixels at 600 dpi scanning resolution) or showed abnormal circularity values (> 1.1 or < 0.85) were excluded from further analysis. Strains for which no consistent fitness estimate could be obtained were also excluded (standard deviation of triplicates greater than standard deviation of all colonies of all strains), which removed two strains from our data set. For the rest, individual corrected colony sizes were averaged and condition-specific resistance/fitness scores were determined by dividing the corrected colony size in the condition of interest by that of the control condition without drug. Signal-to-noise ratios were determined by dividing the mean fitness of the internal controls by their standard deviation. The fraction of unexplained variance was determined by dividing the standard deviation of the internal controls by that of the entire dataset.

For GWAS, phenotype values were transformed to normal shape using the Box–Cox method, mean-centred at zero and normalised to unit variance, using PowerTransformer of *scikit-learn* (Pedregosa *et al*, 2011). Genomic variants were called from published and aligned sequence data (Jeffares *et al*, 2015) using *freebayes* (preprint: Garrison & Marth, 2012), with the following options: –ploidy 1 –standard-filters –min-coverage 10 –min-alternate-count 3. The version of the reference genome used was ASM294v2. SNPs within 3 bp of an indel were filtered out using the –SnpGap option of *bcftools* (Li *et al*, 2009). Low-quality calls and loci where > 50% of the population was not genotyped were removed using the –max-missing 0.5 –minQ 30 –remove-filtered-all options of *vcftools* (Danecek *et al*, 2011). Variant effects were predicted using SnpEff

(Cingolani *et al*, 2012). Variants were filtered for a minor allele frequency of > 5% and converted to plink format using *plink* (Purcell *et al*, 2007). A kinship matrix was constructed in LDAK5 (Speed *et al*, 2012, 2017) by first cutting and thinning predictors, then calculating their weights and finally using the direct method for obtaining the kinship matrix. All steps used default options. Heritability estimates were obtained by REML as implemented in LDAK5. Linear mixed-model association was performed in LDAK5, using the previously generated kinship matrix to correct for population structure.

### Phylogenetics

The phylogenetic tree in Fig 1 was constructed by filtering biallelic SNPs in the region ± 500 bp around the *pyk1* gene (I:1:3,844,243–3,847,145) using *bcftools*. This vcf was converted to a pseudo alignment in fasta format with *VCF-kit* (Cook & Andersen, 2017). The tree was constructed by the neighbour-joining method, implemented in *ClustalW2* (Larkin *et al*, 2007), accessed through the EBI web interface (Goujon *et al*, 2010) and drawn in seaview (Gouy *et al*, 2010).

### Construction of allele replacement strains

The allele replacement strains were generated using the CRISPR-Cas9 system. The plasmid containing the gRNA targeting the *pyk1* SNP in *968 h⁹⁰* (homothallic) and *972 h⁻* (heterothallic) was generated as described (Rodríguez-López *et al*, 2016) using the following primers: gRNA.JB50-F: GCTTTCCGGTGAGACTACCAgttttagagctagaa atagc and gRNA.JB50-R: TGGTAGTCTCACCGGAAAGCttcttcggtac aggttatgt.

Proper gRNA cloning was assessed by Sanger sequencing. The template for homologous recombination was generated using the following primers (underlined is the point mutation introduced in the T-strain to convert it to the A-strain):
HR_JB50-F: ACCCTCGTCCTACTCGTGCCGAGGTTTCCGATGTTGG TAACGCCGTTCTCGATGGTGCTGACTTGGTCATGCTTTCCGGTGAG ACTG<u>C</u>CAAGGGTTCTTA
HR_JB50-R: GTAAGGGATGGAAGCCTCAGCAACACGGGCAGTCT CAGCCATGTAGGTAACGGCTTCAACGGGGTAAGAACCCTTGG<u>C</u>CAG TCTCACCGGAAAGCATGACC

The following primers were used to identify and confirm successful mutants, of which three were kept (referred to as A,B,C in this manuscript) and used for experiments in order to reduce the risk of observing the effects of off-target mutations:
Pyk1ck-F: GATGTTGGTAACGCCGTTCT
Pyk1ck-R: GGACGGTACTTGGAGCAGAG

### Cell culture for multi-omics experiments

Transcriptomes and proteomes were measured from the same cell culture, in five biological repeats per strain. Strains were woken up on yeast extract with supplement (YES) agar and incubated for 2 days at 32°C. Then, 50 ml pre-cultures (YES medium, 32°C, 170 rpm) were grown overnight and used to inoculate 200 ml cultures (YES medium) at an $OD_{600}$ of 0.1. These cells were then grown until $OD_{600}$ of 0.8 and harvested as described below.

## Transcriptomics experiments

When cells reached OD 0.8, 25 ml was collected by centrifugation and snap-frozen in liquid nitrogen. RNA was extracted with a hot phenol method as described in Ref. Lyne *et al* (2003). RNA was further purified with Qiagen RNAeasy columns, and DNAse treatment was performed in the columns (as suggested by manufacturer) prior to library preparation. RNA quality was assessed with a Bioanalyzer instrument (Agilent), and all samples presented a RIN (RNA Integrity Number) > 9. cDNA libraries were prepared with the Illumina TruSeq stranded mRNA kit, according to the manufacturer's specifications, by the Cologne Center for Genomics (CCG) facility. The samples were sequenced on a single lane of an Illumina Hiseq4000 to produce $2 \times 75$ nt reads.

Reads were trimmed with Trimmomatic (Bolger *et al*, 2014) v0.36, with the following parameters differing from default settings: LEADING:0 TRAILING:0 SLIDINGWINDOW:4:15 MINLEN:25. The reference genome was indexed with bowtie2-build with default settings. Paired reads were aligned to the reference genome using bowtie2 with default settings (v2.3.4.1) (Langmead & Salzberg, 2012). In the case of the A-strain, the reference genome was edited to reflect the base substitution within *pyk1*. Aligned reads were counted using *intersect* from the bedtools package (v2.27.1) (Quinlan & Hall, 2010), with the parameters *-wb -f 0.55 -s -bed*. Identical reads were only counted once.

Readcounts were tested for differential expression between strains using DESeq2 v1.18.1, with default settings (Love *et al*, 2014).

## Proteomics experiments

For sample preparation, cells were washed with PBS and centrifuged (3 min, 600 *g*, RT). Subsequently, cells were washed with 1 ml RT lysis buffer (LB: 100 mM HEPES, 1 mM $MgCl_2$, 150 mM KCl, pH 7.5) and transferred to 1.5-ml tubes and centrifuged (5 min, 600 *g*, RT). Cell pellets were flash-frozen and stored at −80°C. Cell pellets were resuspended in 400 µl cold LB, mixed with the same volume of acid-washed glass beads (Sigma-Aldrich), transferred to a FastPrep-24TM 5G Instrument (MP Biomedicals), and disrupted at 4°C by 8 rounds of bead-beating at 30 s with 200 s pauses between the runs. Samples were centrifuged (2 min, 1,000 *g*, 4°C), supernatants collected, and protein concentrations determined with the bicinchoninic acid assay (Thermo Fisher Scientific). Then, 100 µg of proteome samples was subjected to the sample preparation workflow for MS analysis as reported (Piazza *et al*, 2018). Peptide samples were analysed on a Q Exactive HF Orbitrap mass spectrometer (Thermo Fisher Scientific), equipped with a nano-electrospray ion source and a nano-flow LC system (Waters-M-class). For shotgun LC-MS/MS data acquisition (DDA), 1 µl peptide digests from each sample were injected independently at a concentration of 1 µg/µl. MS1 spectra were acquired as described (Piazza *et al*, 2018). One µl peptide digest from the same samples was also measured in data-independent acquisition (DIA) mode on using the DIA settings reported (Piazza *et al*, 2018). The collected DDA spectra were searched against the *S. pombe* fasta database (Clément-Ziza *et al*, 2014), using the Sorcerer™-SEQUEST® database search engine (Thermo Electron) as reported (Piazza *et al*, 2018). For generation of spectral libraries, the DDA spectra were analysed with Proteome

Discoverer 2.2 as described above and imported in the Spectronaut software (version 8, Biognosys AG). DIA-MS targeted data extraction was performed with Spectronaut version 8 (Biognosys AG) with default settings.

Analyses of protein abundance were performed with the MSstat package (Choi *et al*, 2014) using default parameters, unless stated otherwise. Spectronaut output was converted to the input format of MSstats with the *SpectronauttoMSstatsFormat* function. The normalised peak areas were further processed with the function *dataProcess* and *intensity = "NormalizedPeakArea"*. This included $\log_2$ transformation, median normalisation, the summary of fragments to peptides, and the summary of peptides to proteins. The parameter *featureSubset* was set to "all". We used the *groupComparison* function with linear mixed models to compare protein abundances between the replicates for the two strains. The FDRs and log2 foldchanges between strains returned by *groupComparison* were used for further analyses.

## Functional enrichment analyses

Gene ontology (GO) enrichment analysis was performed with the *topGO* package (v2.30.1) (Alexa *et al*, 2006). The annotations were downloaded from PomBase (uploaded on 1st Sept 2015; Wood *et al*, 2012). The transcriptome and proteome were tested separately. All genes with an FDR ≤ 10% were included in the test set, while all other genes formed the background. Importantly, only those genes with available measurements were included in the background to avoid false-positive enrichments. The *nodeSize* was set to 10. We performed Fisher's exact tests with the *elim*-algorithm. All terms with $P \leq 0.01$ were included in the plots. We used AnGeLi (Bitton *et al*, 2015) for functional enrichment analysis to confirm the GO enrichments and to reveal overlap with core environmental stress and TORC1 response genes.

## Metabolomics experiments

Overnight, cell pre-cultures were diluted to $OD_{600}$ of 0.1, and 5 ml was quenched in 20 ml dry-ice-cold methanol when an $OD_{600}$ of 0.8 was reached. This suspension was spun down (600 g, 3 min, 4°C), and the supernatant was discarded by inversion. The pellet was resuspended in the remaining liquid and transferred to a small tube and spun down again with the same parameters. The supernatant was removed completely, and the pellet was frozen in liquid nitrogen and stored at −80°C until further processing.

The samples were extracted as described (Bligh & Dyer, 1959). Acid-washed Zirkonia beads were added to the pellet, together with 140 µl of 10:4 MeOH/water, and cells were lysed mechanically (FastPrep Instrument, 40 s, 6.5 m/s). Then, 50 µl chloroform was added and mixed thoroughly, followed by 50 µl water and 50 µl chloroform. Insoluble components were removed by centrifugation at 5,000 *g* for 10 min. The aqueous phase was recovered and used without further conditioning. One microlitre was injected for LC-MS/MS analysis. The sample was diluted 1:20 for the analysis of free amino acids, except for glutamine which was quantified without dilution.

The compounds were resolved on an Agilent 1290 liquid chromatography system, using a HILIC amide column (Waters BEH Amide, 2.1 × 100 mm, 1.7 µm particle size) with acetonitrile (solvent A) and 100 mM aqueous ammonium carbonate (solvent B) for

gradient elution at a constant flow rate of 0.3 ml/min and column temperature of 35°C. The gradient programme started at 30% B and was kept constant for 3 min before a steady increase to 60% B over 4 min. Solvent B was maintained at 60% for 1 min before returning to initial conditions. The column was washed and equilibrated for 2 min resulting in a total analysis time of 10 min. Compounds were identified by comparing retention time and fragmentation patterns with analytical standards. The samples were analysed by tandem mass spectrometry coupled to an Agilent 6470 triple quadrupole. The sample was acquired using the Agilent dynamic MRM (dMRM) approach with polarity switching. Peak areas were converted to concentrations using external calibration by standard curves and corrected for the optical density of the culture at the time of harvesting.

### Enzyme assays

Pyruvate kinase activity was assayed as described (Gehrig *et al*, 2017). The homothallic A- and T-strains were grown overnight in 3 ml of YES pre-cultures, diluted to OD 0.15, and grown for a further 5 h. The $OD_{600}$ of cultures at the time of sampling was approximately 1. Lysate was prepared by spinning 2 ml of culture (800 g, 3 min, RT), discarding the supernatant, adding a small amount of glass beads and 200 µl of lysis buffer (10 mM Tris at pH 7, 100 mM KCl, 5 mM $MgCl_2$, 1 mM DTT), breaking the cells with a FastPrep instrument (MP Biomedicals), operated at 4°C, three times for 40 s with 1-min breaks in between, spinning at 8,000 g (3 min, 4°C), and transferring the supernatant to a fresh tube kept on ice and used fresh. Reactions with a total volume of 200 µl in a 96-well plate contained 10 mM Tris at pH 7, 100 mM KCl, 5 mM $MgCl_2$, 20 µg L-lactate dehydrogenase from rabbit muscle, 5 mM ADP (all by Sigma-Aldrich), 10 mM PEP (Molekula) and 200 µM NADH (Bioworld). The reaction mix was warmed to 32°C for 1 min and the reaction was started by adding 4 µl of lysate. The absorbance at 340 nm was measured every ~15 s in a Tecan Infinite M200 Pro plate reader set to 32°C. Absorbance values below 0.2 were set to NA. Absorbance values were converted to concentration using the extinction coefficient 6220 $M^{-1}$ $cm^{-1}$ and a path length of 0.5411 cm (calculated from reaction volume and well diameter). The slope of the concentration trace was determined using the linregress function from the scipy python package, background subtracted (reaction mix without lysate), and divided by the OD of the culture at the time of sampling.

### Measurement of growth rates

For each biological replicate, a colony was picked and grown overnight in a 5 ml YES pre-culture. The pre-culture was diluted to $OD_{600}$ 0.2 in 60 ml of fresh YES with 2% glucose and grown for 8 h, with sampling starting at 2 h and every 2 h. The $OD_{600}$ was determined using a spectrophotometer (WPA Biowave CO8000). To determine doubling times, growth data was $log_2$-transformed and a line-fitted, where the doubling time is the inverse of the slope of the fitted line.

### Glucose uptake measurements

After 8 h of growth, media samples were taken, cleared of cells by centrifugation and stored at −80°C until further processing. Glucose concentrations were determined using a commercial, colorimetric kit

(Glucose (HK) Assay Kit, Sigma-Aldrich, catalogue number GAHK20), following the instructions of the manufacturer with the following modifications. A standard curve was prepared from a twofold dilution series of the supplied standard. Samples were diluted 1:20 in water. Then, 10 µl of diluted sample was added to 190 µl of assay buffer in a 96-well plate and incubated for 15 min. The absorption at 340 nm was measured with a plate reader (Tecan Infinite M200 Pro). Values were subtracted by the water blank and converted to concentrations using the standard curve and dilution factor. For each sample, the measured concentration was subtracted from the amount of glucose in fresh media (determined by the same method) and divided by the OD of the culture at the time of sampling.

### Dry weight measurements

We inoculated 100 ml YES cultures from overnight 5 ml YES pre-cultures at an initial $OD_{600}$ of 0.2 and grew them for 24 h. Cultures were centrifuged, washed once in $dH_2O$, dried at 70°C for 48hrs and weighed on a high-resolution balance.

### Oxygen consumption rate measurements

YES cell cultures (100 ml) were grown overnight to an $OD_{600}$ between 1 and 3. A ~25 ml sample was put into a 25-ml Erlenmeyer flask and stirred at 900 rpm using a magnetic stirrer bar. An oxygen probe (Hanna HI 98193), held with a clamp, was inserted into the flask, resulting in it being completely filled with no remaining air inside it, and the flask was sealed with multiple layers of parafilm. The oxygen saturation of the culture was followed over 7 min and recorded every ~1 min. The slope of the concentration trace was determined using the linregress function from the scipy python package and divided by the OD of the culture.

### Carbon source screen

We used Edinburgh minimal medium (EMM) or YES, depending on the nitrogen source in the final assay medium (Moreno *et al*, 1991). Cells were pre-cultured overnight (5 ml), diluted to approximately $OD_{600}$ 0.2 in the morning, grown for 6 h, centrifuged (400 g, 4 min, RT), washed once in EMM without glucose or YES, resuspended and diluted to $OD_{600}$ 0.2 in EMM without glucose or YES. Carbon sources were used at the same molarity as glucose in standard EMM (2% w/w, 111 mM), except for sucrose, maltose and raffinose (where amount was corrected for number of monosaccharides they contain), pectin (where a saturated solution was used), and glutamine (16% w/w due to low solubility). The $OD_{600}$ was recorded in 384-well plates, every 15 min, with short shaking (15 s) before each measurement in a plate reader (Tecan Infinite M200 Pro). Growth curves were smoothed by first applying a median filter of size 5 and a Gaussian filter with sigma = 3. We obtained maximum slopes for each well by fitting all linear regression models for 12 timepoints over the course of the growth curve and retaining the best one.

### Biolog phenotyping screen

The resistance to various chemical compounds was assessed using Biolog Phenotype MicroArray plates PM22, PM23 and PM25. JB50 and *pyk1*-A-allele $h^{90}$ were grown overnight in EMM, diluted to $OD_{600}$

0.15 in fresh EMM and grown for 6 h at 25°C. Cultures were then diluted to $OD_{600}$ 0.05, and 100 µl was added to each well. For each plate type, two individual plates were used (one per strain). Growth curves were recorded by measuring the absorbance at 610 nm every 30 min in an EnVision 2104 plate reader (PerkinElmer) with stacker module. The room was not strictly temperature controlled but was stable at $23.5 \pm 1$°C over the course of the experiment. Growth curves exhibited considerable noise levels and were smoothed by first applying a median filter of size 5 and a Gaussian filter with sigma = 3. We obtained maximum slopes for each well by fitting all linear regression models for 12 timepoints over the course of the growth curve and retaining the best one. Each plate contained multiple concentrations of the same compound, and dose–response curves were plotted for all, with hits identified by manual inspection.

To assess the ability of both strains for using different nitrogen sources (Biolog Phenotype MicroArray plate PM3), we applied the same strategy with the following modifications. We used strains with the heterothallic $h^-$ mating type to prevent mating and sporulation in poor nutrient conditions. Pre-cultures were diluted to $OD_{600}$ 0.2 and grown for 6 h at 25°C. Cultures were then centrifuged (300 g, 3 min, RT), washed in EMM without nitrogen or carbon (EMM-N-C), resuspended in EMM-N at an $OD_{600}$ of 0.2 and 100 µl of cells was added to each well of the assay plates. $OD_{610}$ was recorded in 15-min intervals, and the fit range for maximum slope extraction was accordingly doubled to 24 timepoints. Growth curves were otherwise acquired and analysed similarly. Nitrogen sources in which both strains had a maximum growth rate < 0.015 were excluded from further analysis.

### Spot assays

Three independent pre-cultures were used for the T-strain and one pre-culture per independent CRISPR-engineered mutant. Pre-cultures were grown in either YES or EMM media depending on the plate used for the actual assay. Overnight cultures were diluted to an $OD_{600}$ of 0.15 and grown for an additional ~6 h. Cultures were then diluted to a $OD_{600}$ of 0.4, with a threefold dilution series prepared in a 96-well plate (one culture per column). The "1 to 16 array single source" program of the RoToR HDA (Singer Instruments) was used to create the read-out plates. For each batch, a control plate without toxin was prepared to check for any accidental bias in strain dilutions.

### Chronological lifespan assays

Chronological lifespan assays were performed as previously described (Rallis *et al*, 2013). Single colonies were picked and inoculated in YES. Cells were grown for 48 h, which was treated as the beginning of stationary phase (Day 0). For the A-strain, three independent CRISPR-engineered mutants were used as biological repeats.

## Data availability

The datasets produced in this study have been made available as described below:

- Mass spectrometry proteomics data: PRIDE (Perez-Riverol *et al*, 2019) PXD017833 (http://www.ebi.ac.uk/pride/archive/projects/PXD017833)
- RNAseq data: ArrayExpress E-MTAB-8847 (http://www.ebi.ac.uk/arrayexpress/arrayexpress/experiments/E-MTAB-8847/)
- GWAS phenotypes: Dataset EV1
- GWAS top hits: Dataset EV2
- Metabolomics data: Dataset EV3
- Gene-expression data: Dataset EV4
- Carbon source screen: Dataset EV5
- Nitrogen source screen: Dataset EV6

**Expanded View** for this article is available online.

## Acknowledgements

We are grateful to the Francis Crick Institute Bioinformatics Core facility for advice on phylogenetic trees, Doug Speed for advice on heritability estimation and GWAS, Jamie Macpherson for advice on PYK activity assays and David Ellis for critical reading of the manuscript. Mass spectrometry measurements for proteomics were carried out at the Functional Genomics Center Zurich (FGCZ). This work was supported by a Wellcome Trust Senior Investigator Award (095598/Z/11/Z) to JB. ST was supported by a Boehringer Ingelheim Fonds PhD Fellowship. JG received funding from the DFG (Grants CRC 680 and CRC 1310). The Francis Crick Institute receives its core funding from Cancer Research UK (FC001134), the UK Medical Research Council (FC001134) and the Wellcome Trust (FC001134).

## Author contributions

SK, AB, MR and JB conceived the study. SK, JG, MR-L, SJT, MM, VC, GS and CC-M performed the experiments. SK and JG analysed the data. PP, AB, MR and JB supervised the work and acquired funding. SK and JB drafted the manuscript. All authors read and approved the final submission.

## Conflict of interest

The authors declare that they have no conflict of interest.

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
