## [Review Process File · Molecular Systems Biology]

Pyruvate kinase variant tunes carbon metabolism, growth and stress tolerance in fission yeast

Stephan Kamrad, Jan Grossback, Maria Rodriguez-Lopez, St John Townsend, Michael Mulleder, Valentina Cappelletti, Gorjan Stojanovski, Clara Correia-Melo, Paola Picotti, Andreas Beyer, Markus Ralser and Jürg Bähler.

Review timeline:

Submission date:	26 th September 2019
Editorial Decision:	6 th November 2019
Revision received:	8 th March 2020
Editorial Decision:	11 th March 2020
Revision received:	12 th March 2020
Accepted:	18 th March 2020

Editor: Maria Polychronidou

Transaction Report:

1st Editorial Decision

6th November 2019

Thank you again for submitting your work to Molecular Systems Biology. We have now heard back from the three referees who agreed to evaluate your study. As you will see below, the reviewers acknowledge that study seems interesting. They raise however a series of concerns, which we would ask you to address in a major revision.

The reviewers' recommendations are rather clear and therefore I see no need to repeat the points listed below. Please feel free to contact me in case you would like to discuss in further detail any of the issues brought up by the reviewers.

REFeree REPORTS

Reviewer #1:

The manuscript reports that *S. pombe* with a single-nucleotide change in pyruvate kinase (Pyk1) can switch from aerobic fermentation to respiration. To find this important single nucleotide, Kamrad and colleagues screened a library of *S. pombe* isolates for antimycin A resistance, reasoning that with this drug they can identify strains that do not rely on respiration. Through GWAS, the authors identified the loci linked to resistance and found a T343A amino-acid sequence change in Pyk1. To understand the impact of pyk1 SNP, the authors modified the T-allele in a laboratory strain to a A-allele by CRISPR/Cas9, and then subjected the two strains to all different kinds of high-throughput analysis (RNA seq, proteomics, metabolomics). Through spot assays, they authors further demonstrated that T-allele with low Pyk1 activity was associated with higher resistance to oxidative stress, in line with the higher degree in respiration in this strain.

Overall, this is an interesting study. The content of the story is simple and straightforward; the story

is very well written, and the technical execution of it is excellent. Here, I would like to particularly highlight that I very much appreciated the robustness of the experiments and analyses. Unlike many other current studies, here, the authors generated very solid results with sufficient replicates and controls, overall allowing to draw robust conclusions. I would thank the authors for this very solid and interesting work.

The finding of the authors is interesting, and in line with seems to emerge more and more recently: Yeast (and likely also other organisms) seem to have to completely different operational programs/or metabolic modes, which seem to not being triggered by extracellular conditions, but rather by some intracellular signal, where evidence suggests that this signal might emerge from the rate of glucose uptake or the glycolytic flux. While this view is not completely novel, any further evidence towards it (such as the current study) is crucial because it will get us closer to a true system-level understanding of metabolic (and cellular) functioning. As indicated in the comments below, I hope that the authors can further flash out this point in their discussion. Beyond, I have some more comments that should not be difficult to address.

Major comments

1) Although at some point the authors mentioned that they measured physiological parameters, but I think they did not really do so. With physiological parameters I would understand growth rates, cell dry weight specific-glucose uptake rates, and rates of ethanol (and other by product) excretion, from which one could also calculate yield coefficients (biomass yield, ethanol production yield). While the indicate indicate some growth rates and some oxygen consumption rates, I would consider these only rather very qualitative estimates, and not real physiological data.

Real quantitative measurements of the physiological data are needed to support some statements: (i) on the basis of the increased FBP levels, the authors argue that the glycolytic flux is high. While this reviewer agrees with this inference, it is yet now proof. (ii) Somewhere in the discussion the authors mention something like that less biomass is formed, but without a quantitative estimate on the biomass yield, this remains a rather vague statement. Thus, I would like to ask the authors to add a full and quantitative analyses of the physiological parameters to their otherwise nice data set.

Otherwise, it would be a pity that the generated data set with all the fancy high-throughput techniques would lack the very basis of good old physiological data. Next to completing the data set, I think one would then also be able to say - for sure - whether a change in glycolytic flux occurred, which could (as mentioned above) be THE trigger for the metabolic switch.

2) The data suggest that the in vitro observed increased activity (implemented by the point mutation) leads to an increased glucose uptake rate/glycolytic flux (as inferred by the increased FBP levels, and which should be further validated, see previous point), which in the following leads to a change in metabolism, and in the following to a change in growth rate, and oxygen stress resistance. Then, this would argue that it is the glycolytic flux that is responsible for the shift in the metabolic programs (as others have argued before). Thus, to further test this, it would be excellent if the authors in the strain with the 'engineered' increased Pyk activity could reduce the flux into/through glycolysis, for instance, by replacing hexose transporters (as done by <https://www.ncbi.nlm.nih.gov/pubmed/15345416>) or by limiting glucose concentrations through the use of chemostat-cultures. Here, I would expect that one would obtain again the phenotype of the "low activity pyk allele". I acknowledge that this would mean more work. I think it would be super cool if the authors could do this. Maybe they have already done further work toward this. If they have, it would be awesome if they could include this. If they consider this way over the top, then I am also fine with it.

3) Discussion: Please flash out the meaning of the work more: i.e. that the two different metabolic programs are merely triggered on the basis of some intracellular cues. In this light, I think the authors should include the recent work of Mattanovich et al (<https://www.ncbi.nlm.nih.gov/pubmed/30464212>) where it was shown that the mere overexpression of one transcription factor in *Pichia pastoris* can turn this Crabtree negative yeast into a Crabtree positive one. This work here and the Mattanovich work overall goes along what had been proposed in a perspective a few years ago (cf. <https://www.ncbi.nlm.nih.gov/pubmed/22129078>, Fig. 5). In my point of view, the authors contribute novel evidence to this view and I think the authors should highlight this more.

Minor comments

1) Page 3 (Introduction Part), Paragraph 3, Line 4: '*S. pombe*' should be written in italic style.

2) Page 4, Fig. 1C: it would be better if authors could explain the meaning of the blue and orange boxes in the caption.

3) Page 8, Paragraph 2, Line 11-12: authors indicated that the A-strain had a significantly higher concentration of reduced isoform of L-glutathione (Fig. 2D). However, this information could not be

directly derived from this figure.

4) Page 13, Fig. 4E, F: in the method part, authors indicate that the heterothallic h- strain was used in the biology phenotyping screen to prevent mating and sporulation in poor nutrient conditions. In fourth graph of Fig. 4F was the h90 strain spotted on 8 mM EGTA rather than h-, as suggested by Fig. 4E? Have the authors compared the physiology of h90 and h- for both A and T strains? What is the major difference between h- and h90 strain? Further discussions and clarifications will be needed.

5) Page 18-20, Proteomics experiment: this method part could be shortened in a more concise manner, like the other sections.

6) Do the authors know whether the point mutation affects the allosteric regulation of PYK by FBP? They might want to comment on this.

7) Some statement in the manuscript sounded to me like papers from the cancer metabolism field, where there is always the contrast made between either "biosynthesis" or "energy generation". Personally, I don't think that in the intertwined nature of metabolism such a strict divide is possible. In this light, I would like the authors to reconsider rewriting a number of statements, e.g. the first sentence of the abstract. Here, it sounds like cells do glycolysis for biosynthesis, and respiration for energy (again the view that many colleagues in the cancer metabolism field propose). Yet, (at least in yeast, grown on glucose as the sole carbon and energy source), respiration/TCA supports both energy generation and biosynthesis and also glycolysis generates/supports both energy generation and biosynthesis, and in fact even "feeds" the TCA/respiration. Thus, I feel the contrasting divide as expressed in the first sentence in the abstract is incorrect.

8) In the first paragraph of the introduction, the authors state that with "ample glucose" organisms suppress respiration and TCA cycle (and in the 2nd "limiting glucose conditions"). While this is not wrong, I feel that we by now know much better: It is the rate of glucose uptake/rate of glycolysis that determined whether cells suppress respiration and TCA cycle (cf. <https://www.ncbi.nlm.nih.gov/pubmed/22129078>, <https://www.ncbi.nlm.nih.gov/pubmed/15345416>). In fact, also the authors for shows that it is not the mere glucose levels that determine this, as I guess they grew both the A- and T-allele under high glucose. Thus, please rephrase.

9) Beginning of the result section, you mention "narrow-sense heritability". Could you explain this a bit more to make the MS more accessible to a broader readership.

10) At some point the authors mention Mae2, and its potential function as a gluconeogenic enzyme. Mae2 could also have a function as anaplerotic enzyme, which I consider more likely under the here studied glucose conditions. The authors may consider rephrasing.

11) Fig. 4B: Please indicate in the caption of what the y-axis data mean.

12) Connected with one of the main comments, I feel that the way the author determine the growth rate is not ok (or at least questionable). I agree that determining growth rates is something old of the last century, and omics data are much more fancy. Please determine these old-fashioned things as solidly and thoroughly as you performed the rest of your work. To this end, you might want to get inspired by some older work where they did proper physiological analyses.

Reviewer #2:

In this study, Kamrad et al. investigate how cells balance fermentation and respiration, two processes that are often counter-regulated, across species, to maintain metabolic homeostasis. They first assessed the sensitivity of several *S. pombe* isolates to the respiratory chain inhibitor antimycin A, then used comparative genomics to identify candidate genetic variants that correlated with sensitivity. They found a strong association between sensitivity to antimycin A and a SNP in the glycolytic enzyme pyruvate kinase (PYK). This SNP encodes a threonine (T) at PYK position 343 in sensitive, and alanine (A) in resistant strains. They go on to show that replacing the T PYK variant with the A variant (T343A) in the laboratory strain 972 leads to increased PYK activity, higher glycolytic flux, decreased respiration and, likely due to the latter, increased antimycin A resistance. The authors surveyed the transcriptional and proteomic profiles of the T343A strain and found expression changes that they propose may explain the metabolic shift upon PYK variant switching. Interestingly, T343A-expressing cells proliferate faster than the wild-type strain, however, they are more sensitive to oxidative stress. Together, this study's findings suggest that the emergence of the low activity PYK variant is linked to the trade-off between lower growth rates and increased resistance to oxidative stress.

Overall, this is an interesting study that discusses important questions pertaining to functions of metabolic adaptation and trade-offs between growth rates and stress resistance. The main conclusions of the study are well supported by the experimental data, albeit largely descriptive, e.g. the authors show that the T-to-A PYK variant switch is accompanied by expression changes that resemble expression profiles associated with TORC inhibition and oxidative stress, but do not follow up on these hypotheses. This work's findings provides several testable ideas that can be addressed in future mechanistic studies (i.e. not required for this review), but addressing the following points would strengthen the authors' observations:

1) The authors favour the idea that PYK variant switch is causal (end of paragraph 1 of discussion) to the associated characteristic genome-/proteome-wide expression changes that, in turn, account for the resulting metabolic adaptation.

> Could the authors replace the A with the T allele in strains that naturally express the A variant and report the effects on key metabolic properties they discuss in their study (respiration, glycolytic rate, sensitivity to antimycin A and anti-oxidants, proteome changes)?

2) The authors show that *S. pombe* has only one PYK orthologue with low activity and cannot switch to a high activity isoform like other yeast strains. However, in addition to a genetic PYK isoform switch, it has been demonstrated that PYK is regulated by ultrasensitive allostery that can largely account for the metabolic shift of *S. cerevisiae* upon glucose withdrawal [Xu et al. (2012) *Molecular Cell*, 48:52-62]. Similarly, a fast (a few minutes) rewiring of glycolysis that precedes gene expression changes underlies survival of *S. cerevisiae* upon oxidative stress [Kuehne et al. (2015) *Molecular Cell*, 59:359-371, Ralser, M. (2007) *J. Biol.* 6, 10; Ralser, M. (2009) 27:604-605]. In this manuscript, the PYK T-to-A switch causes global changes to the proteome, and it is impossible from the data presented to disentangle to which extent such fast mechanisms in *S. pombe* might contribute to the differential sensitivity to stress.

> The authors should clarify if *S. pombe* PYK is regulated by allostery or signaling, particularly under the treatment conditions they use in this study, as this is not clear from the manuscript's text and discussion of cited papers.

> How does PYK activity and glycolytic flux compare in the T and A (allele-switched) strains upon acute treatment with antimycin A and oxidative stress?

Other points to be addressed through discussion and changes to the text:

3) It is not clear what conditions may impose the necessary selective pressure for the T variant to arise. Do the authors suggest that the T allele arose as a consequence of oxidative stress in a specific environment/culture conditions?

4) The authors should revisit the use of the term metabolic "reprogramming" (e.g. in abstract): it is unclear from the presented experiments if the observed proteome changes are driven by a specific series of events triggered by PYK allele switch (a possibility that could have been addressed by assessing the acute effects of allele switch on metabolism/ox. stress sensitivity) or are the result of progressive selection of cells with favourable traits. Similarly, the authors should reconsider the use of the term "natural metabolic switch" (end of introduction), which would be more appropriate for mechanisms described in point 2, above.

Reviewer #3:

The authors examined respiration dependence of 161 fission yeast strains. They then performed genome-wide association study (GWAS), and among the top associations, they studied a variant carrying an SNP in *pyk1*, which encodes single pyruvate kinase in lower glycolysis. They then replace the 'low-activity' allele with 'high-activity' allele, and found that SNP in *pyk1* is sufficient to trigger system-level change in metabolism, gene expression, and also cellular traits. They presented comprehensive data (metabolomics, transcriptomics and proteomics), which consistently show increased PYK activity leads to increased glycolysis/fermentation and decreased respiration. This is in line with an earlier report in budding yeast which shows low PYK activity activates yeast respiration (Grüning et al). Overall, it's pretty interesting that this mutation exists in the "wild" and is sufficient to have these somewhat material effects on physiology, and the evidence is well laid out to convey this message.

Major:

Page 3, I suggest explaining the antimycin resistance score in the main text in addition to figure caption.

Page 5, There seem to be many other strong linkages in Fig 1B. Can the authors at least look into these a little more?

Page 8, the authors claim higher glycolytic flux in A-strain (more active pyk) and lower in T-strain (less active pyk), based on empirical evidence that FBP abundance correlates with glycolytic flux. This conclusion could be enhanced by showing side-by-side glucose consumption, oxygen consumption, and waste production.

Page 9, the authors claim lower oxidative burden in A-strain but their evidence are confusing to me. They saw change in NADPH/NADP ratio but no change in ratio between reduced and oxidized form of glutathione.

Page 9, it is interesting that Mae2 (malate/oxaloacetate → pyruvate) is upregulated in both transcription and protein level in T-strain. Given the study was done in rich media with many other nutrients, does it suggest the cells are taking alternative carbon sources into TCA and direct the carbon flow to gluconeogenesis?

General question, how many observed features are related to growth rate and indirectly caused by PYK mutation? For example, comments like "pyk1 SNP differentially affected the resistance to this broad range of stresses, suggesting a general role of glycolysis in stress resistance..." may be more about growth rate

Minor:

1. Page 6, use heat map (and clustering) to present metabolomics data?
2. Many graphs do not go to zero Y-axis. This is okay in selected cases (e.g. energy charge does typically range from 0.7 - 0.9) but should not be the default as it exaggerates effect sizes (e.g. it seems logical to have the oxygen uptake go to zero, and many other cases)
3. Fig.2D is supposed to show glutathione according to page 8?
4. Fig 4, there are a bunch of lines on the figures C,D,E which are inconsistent from panel to panel for unclear reasons.
5. Supplementary Fig.4 comes with wrong caption

Below we address the specific concerns with detailed responses.

Reviewer #1

Overall, this is an interesting study. The content of the story is simple and straightforward; the story is very well written, and the technical execution of it is excellent. Here, I would like to particularly highlight that I very much appreciated the robustness of the experiments and analyses. Unlike many other current studies, here, the authors generated very solid results with sufficient replicates and controls, overall allowing to draw robust conclusions. I would thank the authors for this very solid and interesting work.

The finding of the authors is interesting, and in line with seems to emerge more and more recently: Yeast (and likely also other organisms) seem to have to completely different operational programs/or metabolic modes, which seem to not being triggered by extracellular conditions, but rather by some intracellular signal, where evidence suggests that this signal might emerge from the rate of glucose uptake or the glycolytic flux. While this view is not completely novel, any further evidence towards it (such as the current study) is crucial because it will get us closer to a true system-level understanding of metabolic (and cellular) functioning. As indicated in the comments below, I hope that the authors can further flash out this point in their discussion. Beyond, I have some more comments that should not be difficult to address.

Response: We thank the Reviewer for this thoughtful and overall positive assessment of our work. We have improved the manuscript in several places based on this feedback as outlined below.

Major comments

1) physiological data: glucose uptake, biomass, doubling rate, ethanol

Although at some point the authors mentioned that they measured physiological parameters, but I think they did not really do so. With physiological parameters I would understand growth rates, cell dry weight specific-glucose uptake rates, and rates of ethanol (and other by product) excretion, from which one could also calculate yield coefficients (biomass yield, ethanol production yield). While the indicate indicate some growth rates and some oxygen consumption rates, I would consider these only rather very qualitative estimates, and not real physiological data.

Real quantitative measurements of the physiological data are needed to support some statements: (i) on the basis of the increased FBP levels, the authors argue that the glycolytic flux is high. While this reviewer agrees with this inference, it is yet now proof. (ii) Somewhere in the discussion the authors mention something like that less biomass is formed, but without a quantitative estimate on the biomass yield, this remains a rather vague statement. Thus, I would like to ask the authors to add a full and quantitative analyses of the physiological parameters to their otherwise nice data set. Otherwise, it would be a pity that the generated data set with all the fancy high-throughput techniques would lack the very basis of good old physiological data. Next to completing the data set, I think one would then also be able to say - for sure - whether a change in glycolytic flux occurred, which could (as mentioned above) be THE trigger for the metabolic switch.

Response: We agree and have collected a new, quantitative set of physiological data. These can be found in the new Fig. 4 and a corresponding new paragraph in the Results and Methods

sections. In brief, this new data confirms the growth phenotype, and we have calculated the doubling time by log2 transforming the data fitting a line, where the doubling time is then the inverse of the slope of that line. We have also quantified glucose uptake, which is higher in the high-activity PYK strain as expected. Moreover, dry biomass at the end of the growth curve (24h) is slightly higher for the low-activity PYK strain, as expected for a strain that respire more. 2) *alternative low glycolytic flux*

The data suggest that the in vitro observed increased activity (implemented by the point mutation) leads to an increased glucose uptake rate/glycolytic flux (as inferred by the increased FBP levels, and which should be further validated, see previous point), which in the following leads to a change in metabolism, and in the following to a change in growth rate, and oxygen stress resistance. Then, this would argue that it is the glycolytic flux that is responsible for the shift in the metabolic programs (as others have argued before). Thus, to further test this, it would be excellent if the authors in the strain with the 'engineered' increased Pyk activity could reduce the flux into/through glycolysis, for instance, by replacing hexose transporters (as done by <https://www.ncbi.nlm.nih.gov/pubmed/15345416>) or by limiting glucose concentrations through the use of chemostat-cultures. Here, I would expect that one would obtain again the phenotype of the "low activity pyk allele". I acknowledge that this would mean more work. I think it would be super cool if the authors could do this. Maybe they have already done further work toward this. If they have, it would be awesome if they could include this. If they consider this way over the top, then I am also fine with it.

Reply: This is an interesting point. We have attempted to engineer a natural, high-activity strain to carry the low-activity allele. This approach has not been successful, because the natural strains are much more difficult to transform than the laboratory strain. However, our previous work in budding yeast has started with a high-activity strain engineered to have progressive levels of lower activity (Grüning *et al*, 2011). The growth and oxidative stress phenotype observed there are opposite and therefore supports the idea that glycolytic flux is instrumental. The reviewer specifically suggests using the engineered high-activity PYK strain and to then decrease glycolytic flux by a different means to confirm that low PYK activity/flux is the only effect of the mutation causing the phenotype. While this is also difficult in the *S. pombe* wild-strain within the timeframe of this revision, we conducted similar experiments in two previous papers, where we have reduced the glycolytic flux by introducing mutations in Triosephosphate isomerase in *S. cerevisiae* (Ralsler, J. Biol 2007; Gruening *et al*, Open Biol 2014). In both cases, the introduction of a flux barrier in lower glycolysis resulted in a higher flux of the oxidative PPP and increased oxidant tolerance. We have added this information to the manuscript. The suggestion to use glucose transporter chimeras of different affinities is intriguing, but the problem is that lowering glucose uptake capacity would equally affect glycolysis and the PPP (as both start from glucose 6-phosphate). A lower activity of the glucose transporter is hence not expected to shift the glycolytic flux into the PPP.

3) *Improve discussion*

*Please flash out the meaning of the work more: i.e. that the two different metabolic programs are merely triggered on the basis of some intracellular cues. In this light, I think the authors should include the recent work of Mattanovich *et al* (<https://www.ncbi.nlm.nih.gov/pubmed/30464212>) where it was shown that the mere overexpression of one transcription factor in *Pichia pastoris* can turn this Crabtree negative yeast into a Crabtree positive one. This work here and the Mattanovich work overall goes along what had been proposed in a perspective a few years ago (cf. <https://www.ncbi.nlm.nih.gov/pubmed/22129078>, Fig. 5). In my point of view, the authors contribute novel evidence to this view and I think the authors should highlight this more.*

Response: We agree and have better highlighted this aspect of our results in the Discussion and added the two references suggested by the reviewer. We have also improved the flow and readability of the Discussion.

Minor comments

1) Page 3 (Introduction Part), Paragraph 3, Line 4: '*S. pombe*' should be written in italic style.

Reply: Corrected.

2) Page 4, Fig. 1C: it would be better if authors could explain the meaning of the blue and orange boxes in the caption.

Reply: We agree and have edited the text accordingly.

3) Page 8, Paragraph 2, Line 11-12: authors indicated that the A-strain had a significantly higher concentration of reduced isoform of L-glutathione (Fig. 2D). However, this information could not be directly derived from this figure.

Reply: We apologise that the corresponding graph was missing from the figure. We do indeed see a statistically significantly higher concentration of reduced glutathione in the strain with high PYK activity (this information can be found in Appendix Fig. S3 and Dataset EV3). However, the ratio of reduced to oxidised glutathione is not different between the strains. The data with regard to glutathione is hence not conclusive and we now mention this explicitly in the manuscript.

4) Page 13, Fig. 4E, F: in the method part, authors indicate that the heterothallic h- strain was used in the biology phenotyping screen to prevent mating and sporulation in poor nutrient conditions. In fourth graph of Fig. 4F was the h90 strain spotted on 8 mM EGTA rather than h-, as suggested by Fig. 4E? Have the authors compared the physiology of h90 and h- for both A and T strains? What is the major difference between h- and h90 strain? Further discussions and clarifications will be needed.

Reply: The reason why two mating types were used is solely due to practical decisions. These two isogenic strains only differ in their mating behaviour but not in their vegetative growth or physiology, and are used interchangeably by the *S. pombe* community. Initially, the allele swap strain was constructed in the h90 background for determining growth in glucose media and most stress resistance phenotypes. Later in the project, we needed to do growth assays in poor nutrient conditions and chronological lifespan assays, which is not possible in the h90 strain as it will mate in this condition. We therefore constructed the same allele swap strains in the h-strain which cannot self-mate. No difference is expected in mating-suppressing conditions (i.e. glucose media). We performed several assays with both mating types and have not detected any mating-type specific differences between the two sets of allele swap strains. We have now added this information to the legend of Fig. 5E.

5) Page 18-20, Proteomics experiment: this method part could be shortened in a more concise manner, like the other sections.

Reply: We agree and have now shortened this section considerably and refer to published methods.

6) Do the authors know whether the point mutation affects the allosteric regulation of PYK by FBP? They might want to comment on this.

Reply: Pyruvate kinase has been a model enzyme for the study of allosteric regulation (Macpherson *et al*, 2019). However, *S. pombe* Pyk1 has been studied very little. (Nairn *et al*, 1998) have overexpressed *S. pombe* Pyk1 in *S. cerevisiae* and investigated the purified protein. They find that F-1,6-BP is a less potent allosteric activator than it is for *S. cerevisiae* Pyk1p, but in both cases it causes a decrease in the Hill coefficient. In *S. pombe*, the PEP binding behaviour changes completely from a cooperative behaviour to a Michaelis-Menten behaviour with Hill coefficient 1. F-1,6-BP shows only limited effect on ADP kinetics. For this study, we have performed a range of PYK activity assays using cell lysate. FBP does, as expected, appear to have an activating effect at high PEP substrate concentrations, and this applies to the A- as well as the T-strain (see graph below). We have added this to the supplementary material as Appendix Fig. S4.

The T343A mutation is in a region of the protein that is likely alpha-helical and part of the ADP binding pocket (see area highlighted in pink in Fig. 2 of (Schormann *et al*, 2019)). Together with the enzyme assays above, it is therefore more likely that the mutation directly affects substrate binding, rather than allosteric regulation. We have added this information in the Results section initially describing the locus and a sentence to the 1st paragraph of the Discussion.

7) Some statement in the manuscript sounded to me like papers from the cancer metabolism field, where there is always the contrast made between either "biosynthesis" or "energy generation". Personally, I don't think that in the intertwined nature of metabolism such a strict divide is possible. In this light, I would like the authors to reconsider rewriting a number of statements, e.g. the first sentence of the abstract. Here, it sounds like cells do glycolysis for biosynthesis, and respiration for energy (again the view that many colleagues in the cancer metabolism field propose). Yet, (at least in yeast, grown on glucose as the sole carbon and energy source), respiration/TCA supports both energy generation and biosynthesis and also glycolysis generates/supports both energy generation and biosynthesis, and in fact even "feeds" the TCA/respiration. Thus, I feel the contrasting divide as expressed in the first sentence in the abstract is incorrect.

Reply: We agree in principle and have adjusted relevant statements, including the 1st sentence of the Abstract.

8) *In the first paragraph of the introduction, the authors state that with "ample glucose" organisms suppress respiration and TCA cycle (and in the 2nd "limiting glucose conditions"). While this is not wrong, I feel that we by now know much better: It is the rate of glucose uptake/rate of glycolysis that determined whether cells suppress respiration and TCA cycle (cf. <https://www.ncbi.nlm.nih.gov/pubmed/22129078>, <https://www.ncbi.nlm.nih.gov/pubmed/15345416>). In fact, also the authors for shows that it is not the mere glucose levels that determine this, as I guess they grew both the A- and T-allele under high glucose. Thus, please rephrase.*

Reply: We agree and have rephrased these statements.

9) *Beginning of the result section, you mention "narrow-sense heritability". Could you explain this a bit more to make the MS more accessible to a broader readership.*

Reply: Yes, we have edited the text accordingly to define narrow-sense heritability.

10) *At some point the authors mention Mae2, and its potential function as a gluconeogenic enzyme. Mae2 could also have a function as anaplerotic enzyme, which I consider more likely under the here studied glucose conditions. The authors may consider rephrasing.*

Reply: Thank you for this insight. An earlier study investigating *S. pombe* Mae2 has found that "transcription of mae2 was induced when cells were grown in high concentrations of glucose or under anaerobic conditions" and concluded that "The increased levels of malic enzyme may provide additional pyruvate or assist in maintaining the redox potential under fermentative conditions" (Viljoen *et al*, 1999). While we have not suggested that the T-strain produces glucose from malate in rich media, the pyruvate produced by Mae2 could be used in other ways by the cell (e.g. amino acid biosynthesis). However, we recognise that the reaction in question is reversible and that more recent work that suggests possible anapleurotic roles for malic enzyme (Zelle *et al*, 2011). We have now added this possibility to the manuscript as suggested. However, an anapleurotic role would in our opinion not explain the upregulation of *mae2* specifically in the T-strain which overall grows slower (but which may struggle to produce sufficient pyruvate due to low PYK activity). A tracer experiment to address this question would be a substantial amount of work and would not affect our conclusions.

11) *Fig. 4B: Please indicate in the caption of what the y-axis data mean.*

Reply: This figure was based on data collected with a BioLector microfermentor which measures biomass by light scattering but not in a meaningful unit, therefore the axis only says biomass. We have now collected a new set of physiological measurements as requested in Comment 1 and replaced this figure with one showing OD600 (Fig. 4A).

12) *Connected with one of the main comments, I feel that the way the author determine the growth rate is not ok (or at least questionable). I agree that determining growth rates is something old of the last century, and omics data are much more fancy. Please determine these old-fashioned things as solidly and thoroughly as you performed the rest of your work. To this end, you might want to get inspired by some older work where they did proper physiological analyses.*

Reply: We agree and have re-measured OD600 over 6 hours of exponential growth (new Fig. 4) and calculated doubling times the 'old-fashioned way' (by fitting a line to log-scaled data, where the doubling time equals the inverse of the slope).

Reviewer #2

Overall, this is an interesting study that discusses important questions pertaining to functions of metabolic adaptation and trade-offs between growth rates and stress resistance. The main conclusions of the study are well supported by the experimental data, albeit largely descriptive, e.g. the authors show that the T-to-A PYK variant switch is accompanied by expression changes that resemble expression profiles associated with TORC inhibition and oxidative stress, but do not follow up on these hypotheses. This work's findings provides several testable ideas that can be addressed in future mechanistic studies (i.e. not required for this review), but addressing the following points would strengthen the authors' observations:

Reply: We thank the Reviewer for the careful evaluation of our manuscript and the overall positive assessment. We have addressed the reviewer's specific concerns as outlined below.

1) Inverted allele swap

The authors favour the idea that PYK variant switch is causal (end of paragraph 1 of discussion) to the associated characteristic genome-/proteome-wide expression changes that, in turn, account for the resulting metabolic adaptation.

> Could the authors replace the A with the T allele in strains that naturally express the A variant and report the effects on key metabolic properties they discuss in their study (respiration, glycolytic rate, sensitivity to antimycin A and anti-oxidants, proteome changes)?

Reply: We agree that it would be nice to have a A→T allele-swapped strain. We had tried extensively to construct this strain but were unsuccessful because of difficulties transforming wild *S. pombe* strains. Note that our previous work in budding yeast (Grüning *et al*, 2011) has started with a high-activity strain engineered to have progressive levels of low activity. The growth and oxidative stress phenotype observed there is the inverse and therefore supports our conclusion.

2) Pyk activity under ox stress and antimycin

*The authors show that S. pombe has only one PYK orthologue with low activity and cannot switch to a high activity isoform like other yeast strains. However, in addition to a genetic PYK isoform switch, it has been demonstrated that PYK is regulated by ultrasensitive allostery that can largely account for the metabolic shift of S. cerevisiae upon glucose withdrawal [Xu *et al*. (2012) Molecular Cell, 48:52-62].*

> The authors should clarify if S. pombe PYK is regulated by allostery or signaling, particularly under the treatment conditions they use in this study, as this is not clear from the manuscript's text and discussion of cited papers.

Reply: Please see the reply given above in response to Reviewer 1, comment 6.

*Similarly, a fast (a few minutes) rewiring of glycolysis that precedes gene expression changes underlies survival of S. cerevisiae upon oxidative stress [Kuehne *et al*. (2015) Molecular Cell, 59:359-371, Ralser, M. (2007) J. Biol. 6, 10; Ralser, M. (2009) 27:604-605]. In this manuscript, the PYK T-to-A switch causes global changes to the proteome, and it is impossible from the*

data presented to disentangle to which extent such fast mechanisms in S. pombe might contribute to the differential sensitivity to stress.

How does PYK activity and glycolytic flux compare in the T and A (allele-switched) strains upon acute treatment with antimycin A and oxidative stress?

Reply: While the question is intriguing in theory, the rapid response characteristics of Pyk1 in *S. pombe* are unlikely related to the fast oxidation as observed for GAPDH that is studied in the manuscripts the Reviewer cites. GAPDH is so rapidly oxidized, because it possesses an evolved proton relay (not found in pyruvate kinase), that depends on a specific and unique H₂O₂ binding pocket that is specific for GAPDH and that enables the specific oxidation of the active site cysteine (Peralta et al, Nat Chem Biol, 2015). Almost all other proteins in cells are not directly oxidized by H₂O₂ *in vivo* at physiological levels; they require the peroxiredoxin system and are oxidized indirectly, which occurs at much slower rates (Stoecker et al, Nat Chem Biol. 2018 Feb;14(2):148-155.).

Other points to be addressed through discussion and changes to the text:

3) It is not clear what conditions may impose the necessary selective pressure for the T variant to arise. Do the authors suggest that the T allele arose as a consequence of oxidative stress in a specific environment/culture conditions?

Reply: We have uncovered several conditions in which the T-allele would confer a selective advantage. Oxidative stress tolerance is one advantage observed. Which trait, if any, is selected for in nature is impossible to prove, as we do not know the ecological history of the *S. pombe* strains. While a historic evolutionary role of any specific allele is impossible to prove for certain, the strong phenotype associated with this allele, and its persistence in multiple strains and divergent lineages, is in our opinion a strong indicator of an adaptive role. We have improved the Discussion to highlight this point.

4) The authors should revisit the use of the term metabolic "reprogramming" (e.g. in abstract): it is unclear from the presented experiments if the observed proteome changes are driven by a specific series of events triggered by PYK allele switch (a possibility that could have been addressed by assessing the acute effects of allele switch on metabolism/ox. stress sensitivity) or are the result of progressive selection of cells with favourable traits. Similarly, the authors should reconsider the use of the term "natural metabolic switch" (end of introduction), which would be more appropriate for mechanisms described in point 2, above.

Reply: We thank the Reviewer for suggesting caution with the use of the words 'metabolic reprogramming' and "metabolic switch". We agree they mean different things to different people and have limited the use of these terms in the revised manuscript, including in title and abstract. Our allele-swap experiments show, however, that the metabolic changes are not the consequence of progressive selection of cells with favourable traits. They were induced with the single mutation in the naive, clonal strains, without any further genetic mutations or selection necessary. Moreover, the effects were evident in several independently generated allele-swap strains after each transformation, i.e. after only a few generations.

Reviewer #3:

The authors examined respiration dependence of 161 fission yeast strains. They then performed genome-wide association study (GWAS), and among the top associations, they studied a variant carrying an SNP in pyk1, which encodes single pyruvate kinase in lower glycolysis. They then replace the 'low-activity' allele with 'high-activity' allele, and found that

SNP in pyk1 is sufficient to trigger system-level change in metabolism, gene expression, and also cellular traits. They presented comprehensive data (metabolomics, transcriptomics and proteomics), which consistently show increased PYK activity leads to increased glycolysis/fermentation and decreased respiration. This is in line with an earlier report in budding yeast which shows low PYK activity activates yeast respiration (Grüning et al). Overall, it's pretty interesting that this mutation exists in the "wild" and is sufficient to have these somewhat material effects on physiology, and the evidence is well laid out to convey this message.

Reply: We thank this Reviewer for the helpful feedback. We have extensively edited the text at multiple places as suggested and have collected a new set of physiological data as requested. These changes are described in detail below.

Major:

Page 3, I suggest explaining the antimycin resistance score in the main text in addition to figure caption.

Reply: Yes, we have now briefly defined this score in the main text.

Describe other GWAS hits

Page 5, There seem to be many other strong linkages in Fig 1B. Can the authors at least look into these a little more?

Reply: Details of the top-100 GWAS hits are provided in Dataset EV2. In the main text, we briefly describe six GWAS hits predicted to have moderate or high impact and that are located in annotated genes.

Physiological measurements

Page 8, the authors claim higher glycolytic flux in A-strain (more active pyk) and lower in T-strain (less active pyk), based on empirical evidence that FBP abundance correlates with glycolytic flux. This conclusion could be enhanced by showing side-by-side glucose consumption, oxygen consumption, and waste production.

Reply: We agree and have collected a new, quantitative set of physiological data. These can be found in the new Fig. 4 and a corresponding new paragraph in the Results and Methods sections. In brief, we have also quantified glucose uptake rate, which is higher in the A-strain, as expected, even when normalized to the area under the growth. This new data also confirms the growth phenotypes in terms of OD600 and cell numbers. Finally, we measured dry biomass at the end of the growth curve (24h) which is slightly higher for the T-strain, as expected for a strain that respire more.

Page 9, the authors claim lower oxidative burden in A-strain but their evidence are confusing to me. They saw change in NADPH/NADP ratio but no change in ratio between reduced and oxidized form of glutathione.

Reply: We do indeed see a significantly higher concentration of reduced glutathione in the A-strain (this information was available in Appendix Fig. S3 and Dataset EV3). However the ratio of reduced to oxidised glutathione is not different between the strains, which is expected, however, as non-stressed cells maintain their glutathione pool highly reduced.

Page 9, it is interesting that Mae2 (malate/oxaloacetate -> pyruvate) is upregulated in both transcription and protein level in T-strain. Given the study was done in rich media with many

other nutrients, does it suggest the cells are taking alternative carbon sources into TCA and direct the carbon flow to gluconeogenesis?

Reply: Our gene expression analysis does not answer this question. It could be answered with a time-consuming glutamate tracer experiment. However, as this observation is only peripheral to our main conclusions, we consider this outside the scope of our manuscript. Reviewer #1 made the suggestion that Mae2 could be an anaplerotic enzyme. We consider this interpretation plausible and have updated our discussion accordingly.

> General question, how many observed features are related to growth rate and indirectly caused by PYK mutation? For example, comments like "pyk1 SNP differentially affected the resistance to this broad range of stresses, suggesting a general role of glycolysis in stress resistance..." may be more about growth rate

Reply: This is an important question indeed. We briefly highlight in the Discussion that cellular growth and stress resistance are linked with gene regulation, although cause-effect relationships are not understood. We cite multiple references providing conflicting evidence on these relationships. In principle, it is possible that some of the observed traits are indirectly caused via altered growth rate, although previous studies by us and others show that slow growth is not sufficient to increase stress tolerance (e.g., (Rallis *et al*, 2014)) and that the (evolved) roadblocks in glycolysis explain the increased stress tolerance by increased shift of electron flow into the NADPH and GSH pool, and not by slowing the growth rate of cells (e.g., Ralser *et al*, 2007; Gruening *et al*, 2011; Gruening *et al*, 2014;; Peralta *et al*, 2015). In any case, we do not claim that all observed traits are directly caused by the PYK mutation, even if they are triggered by it.

Minor:

Page 6, use heat map (and clustering) to present metabolomics data?

Reply: We have chosen the PCA plot for the main figure for simplicity and space reasons. We have now additionally added a clustermap of log₂ fold changes versus the T-strain as Appendix Fig. S2 (see also below).

2. Many graphs do not go to zero Y-axis. This is okay in selected cases (e.g. energy charge does typically range from 0.7 - 0.9) but should not be the default as it exaggerates effect sizes (e.g. it seems logical to have the oxygen uptake go to zero, and many other cases)

Reply: We agree with the reviewer that y-axes should start at zero if a comparison of the absolute value is the key message of the figure. If its purpose is to show that there is a difference, or if a zero-value is absurd (e.g. in the case of fitness or resistance), a cut y-axis should be used. We agree that for oxygen uptake, a zeroed y-axis makes more sense and similarly now use zeroed plots to report other physiological parameters in the new Fig. 4.

3. Fig.2D is supposed to show glutathione according to page 8?

Reply: We apologise that the corresponding graph was missing from the figure (it was moved to the supplementary material, and has now been brought into the main figure). We have edited the text and legend accordingly.

4. Fig 4, there are a bunch of lines on the figures C,D,E which are inconsistent from panel to panel for unclear reasons.

Reply: Thank you for pointing this out. We have added a dotted line representing “no effect” in panel D. Data for panel C was recorded in a different way, and we used different cut-offs which are not shown for clarity (since there are no hits).

5. Supplementary Fig.4 comes with wrong caption

Reply: Thank you for spotting this error, which is now corrected.

References

- Grüning N-M, Rinnerthaler M, Bluemlein K, Mülleeder M, Wamelink MMC, Lehrach H, Jakobs C, Breitenbach M & Raiser M (2011) Pyruvate kinase triggers a metabolic feedback loop that controls redox metabolism in respiring cells. *Cell Metab.* **14**: 415–427
- Macpherson JA, Theisen A, Masino L, Fets L, Driscoll PC, Encheva V, Snijders AP, Martin SR, Kleijung J, Barran PE, Fraternali F & Anastasiou D (2019) Functional cross-talk between allosteric effects of activating and inhibiting ligands underlies PKM2 regulation. *Elife* **8**: Available at: <http://dx.doi.org/10.7554/eLife.45068>
- Nairn J, Duncan D, Gray LM, Urquhart G, Binnie M, Byron O, Fothergill-Gilmore LA & Price NC (1998) Purification and characterization of pyruvate kinase from *Schizosaccharomyces pombe*: evidence for an unusual quaternary structure. *Protein Expr. Purif.* **14**: 247–253
- Rallis C, López-Maury L, Georgescu T, Pancaldi V & Bähler J (2014) Systematic screen for mutants resistant to TORC1 inhibition in fission yeast reveals genes involved in cellular ageing and growth. *Biol. Open* **3**: 161–171
- Schormann N, Hayden KL, Lee P, Banerjee S & Chattopadhyay D (2019) An overview of structure, function, and regulation of pyruvate kinases. *Protein Sci.* **28**: 1771–1784
- Viljoen M, Volschenk H, Young RA & van Vuuren HJ (1999) Transcriptional regulation of the *Schizosaccharomyces pombe* malic enzyme gene, *mae2*. *J. Biol. Chem.* **274**: 9969–9975
- Zelle RM, Harrison JC, Pronk JT & van Maris AJA (2011) Anaplerotic role for cytosolic malic enzyme in engineered *Saccharomyces cerevisiae* strains. *Appl. Environ. Microbiol.* **77**: 732–738

Thank you again for sending us your revised manuscript. We have now heard back from reviewer #1 who was asked to evaluate your study. As you will see below, the reviewer acknowledges that most previously raised issues have been satisfactorily addressed. Reviewer #1 still raises one remaining issue, which we would ask you to address in a minor revision.

REFEREE REPORTS

Reviewer #1:

The authors nicely addressed my earlier suggestions.

Yet, I suggest that the authors give the following points a thought:

I think the authors have not determined the glucose uptake RATE. It seems that they report "glucose consumed per final OD". From my point of view, this is not a rate, but rather a yield coefficient. The inverted ratio (i.e. final OD per glucose consumed) would be the "biomass yield" coefficient. Here, the A-strain would have a lower biomass yield (1/2.6) vs. (1/2.05) in the T-strain. This would be consistent with a more fermentative metabolism in the A-strain. Yet, please note that the measurements you show in Fig. 4C are not glucose uptake RATES, as there is no "time information" in the unit of your measurement. Thus, the sentence "Accordingly, we detected an increased glucose uptake in the A-strain during 8 hrs of growth, normalized to the final OD (Fig. 4C, $p=0.04$, Welch's t-test)." would need to be adjusted as the measurement as reported is not a rate. Yet, I am still wondering: On the basis of the new measurements (of glucose would have been measured at different time points of the culture), I guess the authors should still be able to determine true glucose uptake rates...

A few sentences later, the authors state "final biomass". I fear that measuring the biomass amount after 24 hours can be somewhat problematic. At this point, the cultures likely went through several post-diauxic phase (as the authors use rich medium) and are now likely stationary. Thus, the reported measurements contain the complete "history" (including the post-diauxic, stationary phase; which might be different for both strains), and therefore it is somewhat questionable whether the "final biomass" amounts are actually informative for what happens in the exponential phase of the culture, i.e. how efficient the metabolism runs in this phase. Thus, I would at least put some question marks on these data.

The orange/blue color scheme is not fully consistent in Fig. 4.

2nd Revision - authors' response

12th March 2020

We agree with the points raised by Reviewer 1 and have edited the manuscript accordingly. Specific responses to the points raised by Reviewer 1 are outlined below.

I think the authors have not determined the glucose uptake RATE. It seems that they report "glucose consumed per final OD". From my point of view, this is not a rate, but rather a yield coefficient. The inverted ratio (i.e. final OD per glucose consumed) would be the "biomass yield" coefficient. Here, the A-strain would have a lower biomass yield (1/2.6) vs. (1/2.05) in the T-strain. This would be consistent with a more fermentative metabolism in the A-strain. Yet, please note that the measurements you show in Fig. 4C are not glucose uptake RATES, as there is no "time information" in the unit of your measurement. Thus, the sentence "Accordingly, we detected an increased glucose uptake in the A-strain during 8 hrs of growth, normalized to the final OD (Fig. 4C, $p=0.04$, Welch's t-test)." would need to be adjusted as the measurement as reported is not a rate. Yet, I am still wondering: On the basis of the new measurements (of glucose would have been measured at different time points of the culture), I guess the authors should still be able to determine true glucose uptake rates...

Thank you for this observation. We agree that our measurements are not rates in the strict sense and have clarified this in the manuscript text (p. 8). We actually only used the word 'rate' in the rebuttal

letter but not in the revised manuscript.

A few sentences later, the authors state "final biomass". I fear that measuring the biomass amount after 24 hours can be somewhat problematic. At this point, the cultures likely went through several post-diauxic phase (as the authors use rich medium) and are now likely stationary. Thus, the reported measurements contain the complete "history" (including the post-diauxic, stationary phase; which might be different for both strains), and therefore it is somewhat questionable whether the "final biomass" amounts are actually informative for what happens in the exponential phase of the culture, i.e. how efficient the metabolism runs in this phase. Thus, I would at least put some question marks on these data.

Unlike budding yeast, fission yeast actually does not feature a distinct post-diauxic growth phase. Nevertheless, we agree with this potential issue which we now mention in the text (p. 8).

The orange/blue color scheme is not fully consistent in Fig. 4.
Thank you for spotting this error, it has been corrected.

Accepted

18th March 2020

Thank you again for sending us your revised manuscript and for performing the last requested text edits. We are now satisfied with the modifications made and I am pleased to inform you that your paper has been accepted for publication.

Corresponding Author Name: Jurg Bahler

Manuscript Number: MSB-19-9270